# *Cis*-regulatory basis of sister cell type divergence in the vertebrate retina

Daniel P Murphy, Andrew EO Hughes, Karen A Lawrence, Connie A Myers, Joseph C Corbo*

Department of Pathology and Immunology, Washington University School of Medicine, St. Louis, United States

**Abstract** Multicellular organisms evolved via repeated functional divergence of transcriptionally related sister cell types, but the mechanisms underlying sister cell type divergence are not well understood. Here, we study a canonical pair of sister cell types, retinal photoreceptors and bipolar cells, to identify the key *cis*-regulatory features that distinguish them. By comparing open chromatin maps and transcriptomic profiles, we found that while photoreceptor and bipolar cells have divergent transcriptomes, they share remarkably similar *cis*-regulatory grammars, marked by enrichment of K50 homeodomain binding sites. However, cell class-specific enhancers are distinguished by enrichment of E-box motifs in bipolar cells, and Q50 homeodomain motifs in photoreceptors. We show that converting K50 motifs to Q50 motifs represses reporter expression in bipolar cells, while photoreceptor expression is maintained. These findings suggest that partitioning of Q50 motifs within cell type-specific *cis*-regulatory elements was a critical step in the evolutionary divergence of the bipolar transcriptome from that of photoreceptors.
DOI: https://doi.org/10.7554/eLife.48216.001

*For correspondence:
jcorbo@wustl.edu

Competing interests: The authors declare that no competing interests exist.

## Introduction

Complex tissues require the coordinated activity of a wide array of specialized cell types. It has been proposed that cellular diversity arises in the course of evolution through a 'division of labor' process, in which a multifunctional ancestral cell type gives rise to descendant cell types with divergent and novel functions (*Arendt et al., 2009*; *Brunet and King, 2017*). Such descendants are often referred to as 'sister' cell types, and typically share a range of morphological, functional, and transcriptional features while at the same time displaying key differences (*Arendt, 2003*; *Arendt et al., 2016*). A canonical example of sister cell types are mammalian retinal photoreceptors and bipolar cells (BCs) (*Arendt, 2008*; *Lamb, 2013*). In a typical vertebrate retina, photoreceptors synapse onto bipolar cells, which, in turn, synapse onto retinal ganglion cells that send their axons to the brain. Bipolar cells therefore constitute the central interneuronal cell class in the vertebrate retina. In mice, an array of 15 distinct bipolar cell types, broadly categorized as ON and OFF based on their response to light onset/offset, serve as a scaffold upon which the complex information-processing circuitry of the retina is built (*Euler et al., 2014*). In this paper, we refer to photoreceptors and bipolar cells as cell 'classes', since they each comprise multiple distinct cell types.

During retinal development, photoreceptors and bipolar cells arise from the same population of OTX2-expressing progenitor cells (*Koike et al., 2007*; *Wang et al., 2014*; *Emerson et al., 2013*), share a similar elongate morphology (*Lamb, 2013*), and possess the molecular machinery required for ribbon synapse formation, a structure not found in any other retinal cell class (*tom Dieck and Brandstätter, 2006*). In some vertebrate species, a subset of bipolar cells exhibit additional photoreceptor-like features, including localization of their cell bodies in the outer nuclear layer and the presence of an inner segment-like structure known as Landolt's club, which extends from the dendrite to the outer limiting membrane and contains a 9+0 cilium (*Tauchi, 1990*; *Hendrickson, 1966*;

**eLife digest** Humans see the world through a light-sensitive tissue at the back of the eye called the retina, which is made up of three layers that each contain specific cell types. The layers form a circuit, with light-sensing photoreceptor cells in the outermost layer connected to bipolar cells in the middle layer, which connect to the brain via specialized cells in the innermost layer. Photoreceptors and bipolar cells share similar characteristics and are thought to be 'sister cells' which evolved from a common ancestral cell type. However, it is not well understood how these two cells types diverged during evolution.

Every cell type has a specific role, which is largely determined by the set of genes that it switches on or off. Specialized regions of DNA, called enhancers, determine whether a gene is turned on or off in a particular cell type. In each cell, DNA strands are bundled together with proteins into a coiled structure known as chromatin. In some cells, a particular enhancer may be 'shut down' and rendered inactive on account of being tightly packed within chromatin. Whilst in other cells, the same enhancer may be 'open' and ready for action. For a given cell type, which genes are turned on is determined, in part, by which enhancers are open.

One way to distinguish between cells is by examining how their chromatin is packaged to see which enhancers are open. Researchers have previously characterized the chromatin structure of photoreceptor cells, but the structure of chromatin in bipolar cells, and how it compares to that of photoreceptors, remained unknown. Now, Murphy et al. have examined the chromatin profile of bipolar cells from the mouse retina in order to gain a better understanding of how these two cell types may be evolutionarily related.

The analysis revealed that although bipolar and photoreceptor cells switch on different sets of genes, the enhancers open in each cell type are very similar. Despite this similarity, Murphy et al. were able to detect subtle differences in short sequences of DNA, known as motifs, present in bipolar and photoreceptor enhancers. Further experiments showed that one of these motifs may be responsible for turning photoreceptor genes off in bipolar cells. This motif therefore appears to play a critical role in distinguishing photoreceptors from bipolar cells.

This comparison of photoreceptor and bipolar cells has provided a possible mechanism whereby photoreceptor and bipolar cells diverged in evolution from a single common ancestral cell type. This insight may help explain how complex organisms with many cell types may have evolved from a single-cell ancestor long ago.

DOI: https://doi.org/10.7554/eLife.48216.002

*Locket, 1970*; *Quesada and Génis-Gálvez, 1985*). These 'transitional' cell types point to the evolutionary origin of bipolar cells from photoreceptors (*Arendt, 2008*; *Lamb, 2013*).

Both shared and divergent features of sister cell types are mediated by the transcriptional regulatory networks that govern gene expression in each cell type. In vertebrates, photoreceptors and bipolar cells express the paired-type homeodomain (HD) TFs CRX and OTX2, which are master regulators of gene expression in both cell classes (*Omori et al., 2011*; *Nishida et al., 2003*; *Hennig et al., 2008*; *Corbo et al., 2010*). A third paired-type HD TF, VSX2, is expressed specifically in bipolar cells and is required for bipolar fate (*Livne-Bar et al., 2006*; *Liu et al., 1994*). Paired-type homeodomains recognize a core 'TAAT' motif, with additional specificity conferred by amino acids in positions 47, 50, and 54 of the homeodomain (*Treisman et al., 1989*; *Noyes et al., 2008*; *Berger et al., 2008*) In particular, a lysine at position 50 (K50, as found in CRX and OTX2) favors recognition of TAAT$\underline{CC}$, whereas a glutamine (Q50, as found in VSX2) favors recognition of TAAT$\underline{T}^A/_G$. Thus, substitution of a single amino acid in the HD toggles the TF's binding preference for the nucleotides 3' of the TAAT core (*Treisman et al., 1989*). Various bHLH TFs, which recognize E-box motifs (CANNTG), also play important roles in photoreceptor and bipolar cell gene expression programs (*Tomita et al., 2000*; *Akagi et al., 2004*). For example, NEUROD1 is required for photoreceptor survival, and BHLHE22 and BHLHE23 are required for development of OFF cone bipolar cells and rod bipolar cells, respectively (*Bramblett et al., 2004*; *Feng et al., 2006*; *Huang et al., 2014*; *Pennesi et al., 2003*).

Our lab has previously shown that the *cis*-regulatory elements (CREs; i.e., enhancers and promoters) of mouse rods and cones are strongly enriched for K50 HD motifs as well as moderately enriched for Q50 HD and E-box motifs (*Hughes et al., 2017*). In addition, we recently used a massively parallel reporter assay to analyze the activity of thousands of photoreceptor enhancers identified by CRX ChIP-seq and found that both K50 HD and E-box motifs are positively correlated with enhancer activity in photoreceptors while Q50 HD motifs have a weakly negative correlation with enhancer activity (*Hughes et al., 2018*). In contrast, studies of individual reporters have shown that Q50 HD motifs mediate weak activation of expression via RAX, and that RAX can either enhance or suppress the transactivation activity of CRX, depending upon RAX expression levels (*Irie et al., 2015*; *Kimura et al., 2000*). Thus, Q50 HD motifs appear to have both positive and negative effects of photoreceptor enhancer activity, depending on context. In contrast, Q50 motifs in bipolar cells appear to be strongly repressive when bound by VSX2, which has been proposed to inhibit the expression of photoreceptor genes in bipolar cells (*Livne-Bar et al., 2006*; *Dorval et al., 2006*; *Clark et al., 2008*). The opposing effects on transcriptional activity mediated by K50 and Q50 motifs suggest that even subtle changes in HD binding sites may mediate major differences in gene expression. Indeed, a recent study in *Drosophila* showed that single base pair substitutions that interconvert Q50 and K50 half-sites within dimeric motifs are sufficient to switch the specificity of opsin expression within photoreceptor sub-types (*Rister et al., 2015*).

Photoreceptors and bipolar cells offer an attractive model system in which to examine the mechanisms of *cis*-regulatory divergence in evolution and development, but the *cis*-regulatory landscape of bipolar cells is currently unknown. To elucidate the *cis*-regulatory grammar of bipolar cells we isolated specific bipolar cell populations from mouse retina and obtained profiles of open chromatin and gene expression. By comparing these datasets to matching data from mouse rod and cone photoreceptors we identified differential enrichment of Q50 motifs in photoreceptor-specific enhancers and a corresponding enrichment of E-boxes in bipolar-specific enhancers. We propose that the differential partitioning of Q50 motifs in photoreceptor and bipolar enhancers was a key evolutionary innovation contributing to transcriptomic divergence between the two cell classes.

## Results

### Photoreceptors and bipolar cells exhibit divergent transcriptional profiles

To obtain cell class-specific transcriptome profiles of mouse bipolar cells, we used fluorescence-activated cell sorting (FACS) to purify bipolar cell populations from adult mice. We first isolated all bipolar cells using *Otx2*-GFP mice. This line harbors a GFP cassette knocked into the endogenous *Otx2* locus (*Fossat et al., 2007*). Adult *Otx2*-GFP mice display high-level GFP in bipolar cells and low-level expression in photoreceptors (*Figure 1B*). To purify ON and OFF bipolar cells separately, we crossed *Otx2*-GFP mice with a *Grm6*-YFP line, in which YFP is driven by the *Grm6* promoter and expressed only in ON bipolar cells (*Morgan et al., 2011*). In the double transgenic mice (*Otx2*-GFP⁺; *Grm6*-YFP⁺), ON bipolar cells co-express GFP and YFP and can be separated from OFF bipolar cells based on fluorescence intensity (*Figure 1B*). We subjected OFF bipolar cells to a second round of sorting to maximize purity from the adjacent weakly fluorescent photoreceptor population. Purity of bipolar cell populations was confirmed by RT-qPCR which showed enrichment of ON- and OFF-specific genes in the appropriate populations and depletion of markers for other retinal cell classes as compared to whole retina (*Figure 1C* and *Figure 1—figure supplement 2*). We then used RNA-seq to profile gene expression in purified populations of bipolar cells, obtaining highly reproducible profiles across biological replicates (*Figure 1—figure supplement 1*).

To define the transcriptional differences between photoreceptor and bipolar cells we compared RNA-seq data from bipolar cells to similar data from wild-type rods and *Nrl*⁻/⁻ photoreceptors previously generated in our lab (*Hughes et al., 2017*). We used *Nrl*⁻/⁻ photoreceptors as a surrogate for blue cones (i.e., *Opn1sw*-expressing cones) since mouse photoreceptors lacking *Nrl* transdifferentiate into blue cones during development (*Daniele et al., 2005*; *Nikonov et al., 2005*). We identified a total of 5259 genes with at least a two-fold difference in expression between bipolar cells and either rods or blue cones (FDR < 0.05) (*Supplementary file 5*). Despite the large number of differentially expressed genes, published single-cell RNA-seq profiles indicate that the bipolar cell

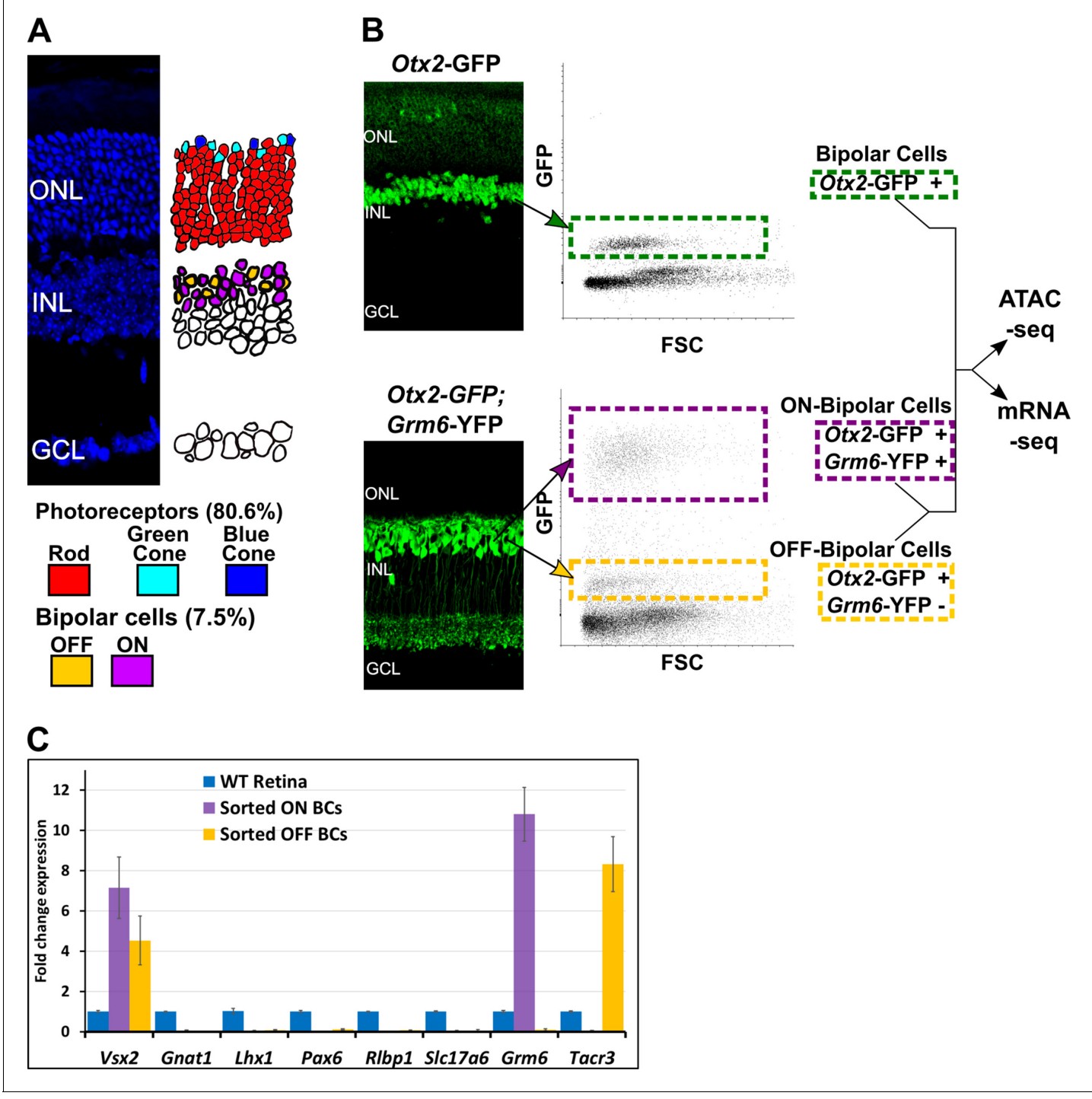

**Figure 1.** Isolation of bipolar cell populations from adult mouse retina. (**A**) Left, Histological section of adult mouse retina stained with 4′,6-diamidino-2-phenylindole (DAPI), to highlight nuclei. ONL = outer nuclear layer, INL = inner nuclear layer, GCL = ganglion cell layer. Right, Schematic depiction of the location and relative abundance of photoreceptor and bipolar cell types. Percentage representation for each cell population in the mouse retina is shown, based on *Jeon et al. (1998)*. (**B**) Left, Histologic sections of retina from transgenic mice expressing fluorescent marker proteins. In *Otx2*-GFP mice, GFP is strongly expressed in all bipolar cells, and weakly in photoreceptors. In *Grm6*-YFP mice, YFP is expressed exclusively in ON bipolar cells. Bipolar cell populations in the INL are linked to their position on FACS scatterplots with arrows (FSC = Forward Scatter). (**Top**). All bipolar cells from *Otx2*-GFP+ mice are boxed in green. Bottom: in *Otx2*-GFP+;*Grm6*-YFP+ mice, ON bipolar cells (purple box) are separable from OFF bipolar cells (gold box) based on intensity of fluorescence. (**C**) RT-qPCR analysis of retinal cell class markers (*Macosko et al., 2015*) in sorted ON (purple) and OFF (gold) bipolar cells normalized to expression in whole retina (blue). *Vsx2* = bipolar cells; *Gnat1* = rod photoreceptors; *Lhx1* = horizontal cells; *Pax6* = amacrine, ganglion, horizontal cells; *Rlbp1* = Müller glia; *Slc17a6* = ganglion cells; *Grm6* = ON bipolar cells; *Tacr3* = OFF bipolar cells (Types 1A, 1B, and 2).

*Figure 1 continued on next page*

*Figure 1 continued*

DOI: https://doi.org/10.7554/eLife.48216.003

The following figure supplements are available for figure 1:

**Figure supplement 1.** Reproducibility of ATAC-seq and RNA-seq datasets.

DOI: https://doi.org/10.7554/eLife.48216.004

**Figure supplement 2.** BC5D is likely excluded from both ON and OFF populations.

DOI: https://doi.org/10.7554/eLife.48216.005

transcriptome is more similar to that of photoreceptors than to that of any other retinal cell class (*Macosko et al., 2015*).

To evaluate the functional differences between photoreceptor and bipolar cell gene expression programs we compared the top ~ 30% most differentially expressed genes in each cell class (832 bipolar cell, 818 photoreceptor) using the gene ontology (GO) analysis tool, PANTHER (*Thomas et al., 2003*). Top bipolar-enriched GO terms were typical of many neuronal cell types and related to aspects of synaptic transmission, while photoreceptor-enriched GO terms mainly related to light-sensing (*Supplementary file 6*). Next, we compared the list of differentially expressed genes to a database of mouse TFs (AnimalTFDB3.0; *Hu et al., 2019*), which revealed that 394 of the differentially expressed genes encode putative transcriptional regulators (*Supplementary file 5*). These include TFs known to be responsible for controlling gene expression in rods (*Nrl, Nr2e3, Neurod1*), cones (*Thrb*), bipolar cells (*Vsx2, Neurod4*), or both cell classes (*Crx*). Nearly one-third (176) of differentially expressed TFs are members of the zinc finger (ZF) family, many of which are more highly expressed in rods compared to bipolar cells but not in blue cone compared to bipolar cells. Conversely, of the top 10% differentially expressed TFs, the majority (25 of 35) are more highly expressed in bipolar cells compared to either rods or cones, and of these, most (16 of 25) are classified as HD, ZF or bHLH. Thus, the transcriptomes of bipolar cells and photoreceptors are significantly divergent, despite various functional and morphological similarities between the two cell classes.

In contrast, comparison of the transcriptomes of ON and OFF bipolar cells identified only 680 genes that were differentially expressed by at least two-fold (317 ON- and 363 OFF-enriched at FDR < 0.05; *Supplementary file 5*). This figure is less than half of the number of differentially expressed genes identified between rods and blue cones (1,471), indicating that the transcriptomes of the two categories of bipolar cells are quite similar.

Of note, a recent study by *Shekhar et al. (2016)*, described single-cell expression profiles for bipolar cell types using Drop-seq. To compare our results with this study, we pooled data from Shekhar et al. across inferred cell types to generate 'pseudo-bulk' ON, OFF, and pan bipolar cell gene expression profiles. We found that expression estimates from bulk and pooled pseudo-bulk single-cell RNA-seq are well correlated (Pearson correlation coefficients range from 0.82 to 0.85, *Figure 2—figure supplement 1A–C*). Similarly, pairwise comparisons showed that bulk ON, OFF, and pan bipolar cell populations were most strongly correlated with their pseudo-bulk counterparts, suggesting that the two approaches yield similar cell type-specific expression profiles (*Figure 2—figure supplement 1D*).

Shekhar et al. also identified an ON bipolar cell type (BC5D) that expresses low levels of *Grm6* and could thus have been incorrectly sorted into our OFF population. To evaluate this possibility, we examined RNA-seq data for *Lrrtm1*, a gene specific to BC5D bipolar cells (*Shekhar et al., 2016*). We observed low read counts for *Lrrtm1* in both ON and OFF bipolar populations, suggesting that BC5D cells were not sorted into either group. To investigate further, we repeated the FACS experiments, this time also collecting cells exhibiting fluorescence levels in between those of the ON and OFF populations (termed YFP-low). We found that the YFP-low population expressed both the ON BC marker *Isl1* and the BC5D marker *Lrrtm1*, while the ON and OFF populations did not express *Lrrtm1* (*Figure 1—figure supplement 2*). These results indicate that low levels of *Grm6*-YFP expression caused BC5D cells to be excluded from both the ON and OFF bipolar populations. Given the rarity of BC5D cells, which constituted 2.3% of cells identified as bipolar cells by Shekhar et al., we believe their absence does not materially impact our analysis.

To gain insight into gene expression among individual bipolar cell types, we compared the list of genes differentially expressed between ON and OFF bipolar cells with the data of Shekhar et al. Overall, we found a strong correlation between the results of bulk ON and OFF bipolar cell expression profiling and single-cell transcriptome analysis (*Figure 2*; *Figure 2—figure supplement 1E* and *Figure 2—source data 1*). Additionally, 50 of the 680 differentially expressed genes found in our bulk analysis were not present in the Drop-seq data (*Supplementary file 5*). These 50 genes are

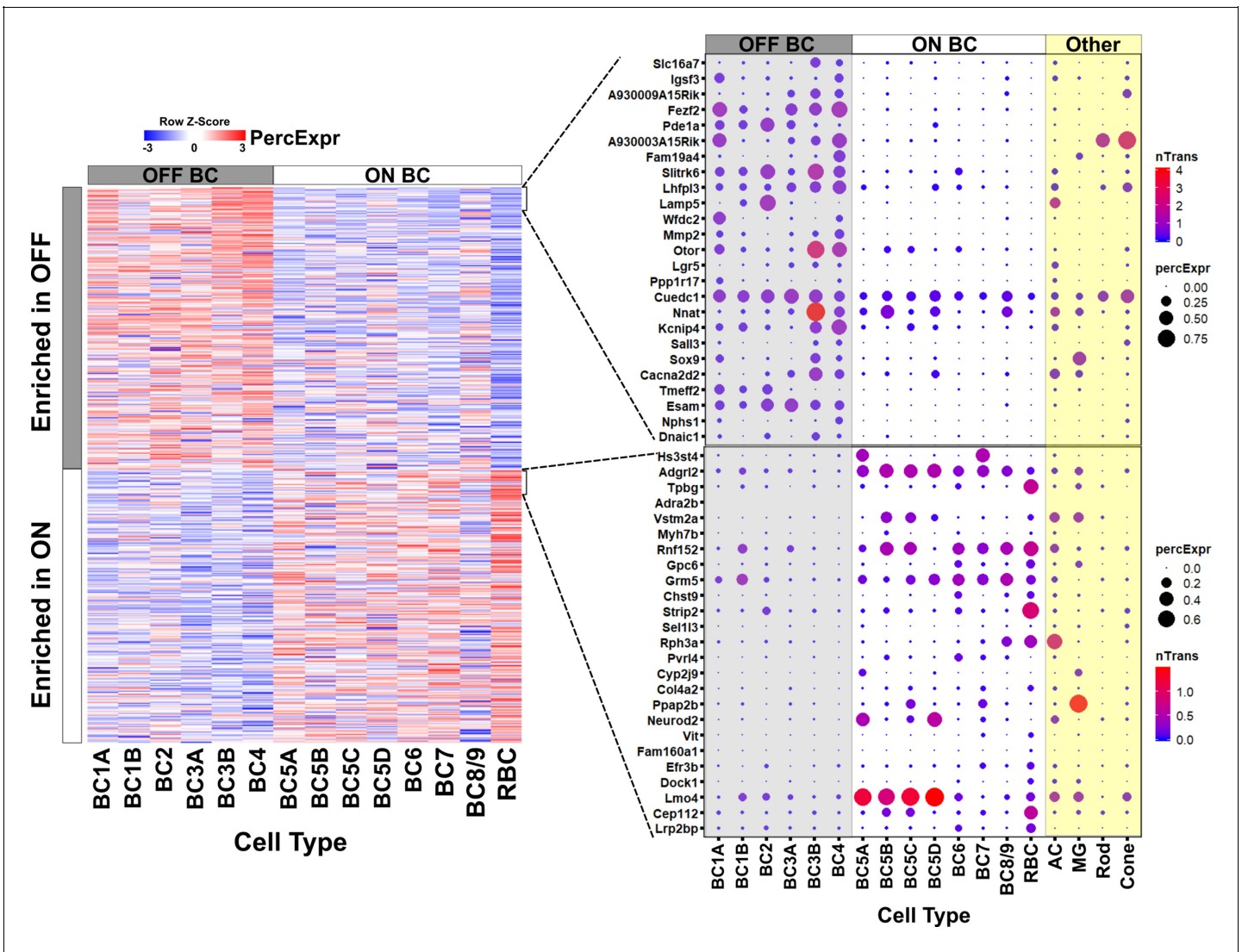

**Figure 2.** Gene expression in ON and OFF bipolar cells. Left, heatmap displaying 680 genes identified by bulk RNA-seq as differentially expressed between FACS-purified ON and OFF bipolar cell populations (current study) mapped onto single cell expression profiles for bipolar cell types identified by Drop-seq (*Shekhar et al., 2016*). Overall, genes identified as ON- or OFF-specific by bulk RNA-seq showed corresponding differential expression between ON and OFF bipolar types identified by Drop-seq. Right, Expression of the top 25 most differentially enriched genes (ranked by p-value) in OFF (top) and ON (bottom) bipolar populations presented as dot plots as in Shekhar et al. nTrans = mean number of transcripts expressed per cell in each cluster identified as a bipolar cell type; PercExpr = percentage of cells within each cluster found to express the indicated gene. Dot plots for all 680 differentially expressed genes are presented in *Figure 2—source data 1*.

DOI: https://doi.org/10.7554/eLife.48216.006

The following source data and figure supplement are available for figure 2:

**Source data 1.** Single-cell expression profiles of genes differentially expressed between ON and OFF bipolar cells.
DOI: https://doi.org/10.7554/eLife.48216.008

**Figure supplement 1.** Comparison of gene expression measured by bulk RNA-seq and single-cell RNA-seq.
DOI: https://doi.org/10.7554/eLife.48216.007

generally expressed at low levels, even in the cell population in which they are enriched. This finding suggests that bulk RNA-seq of purified populations can provide information complementary to that obtained by single-cell profiling.

Taken together, these data indicate that despite their sister cell type relationship, photoreceptor and bipolar cells have markedly distinct transcriptional profiles, while ON and OFF bipolar cells are more similar at the transcriptome level than rods and cones.

## Bipolar cells have a more accessible chromatin landscape than either rods or cones

To compare chromatin accessibility between photoreceptor and bipolar cells, we used ATAC-seq (Assay for Transposase-Accessible Chromatin by sequencing) to generate open chromatin profiles from FACS-purified bipolar cells (*Buenrostro et al., 2013*). Similar to our RNA-seq data, ATAC-seq generated highly reproducible profiles across biological replicates (Pearson correlation 0.95–1.00, *Figure 1—figure supplement 1*). For the purpose of our bioinformatic analyses, we define 'promoters' as those ATAC-seq peaks that occur between 1000 bp upstream and 100 bp downstream of a transcription start site (TSS); we refer to ATAC-seq peaks outside of this range as 'enhancers' or 'TSS-distal' elements. To compare chromatin accessibility across tissues, we combined ATAC-seq peaks from bipolar cells with previously generated ATAC-seq data from purified mouse rods, blue cones, and 'green' cones (i.e., *Opn1mw*-expressing cones) as well as DNase-seq data from whole retina, brain, heart, and liver to obtain a list of > 345,000 open chromatin regions (*Hughes et al., 2017*; *ENCODE Project Consortium, 2012*). Hierarchical clustering of chromatin accessibility profiles at enhancers showed that photoreceptors, bipolar cells, and whole retina cluster separately from other tissues (*Figure 3A*). Thus, the sister cell type relationship between photoreceptors and bipolar cells is reflected by the similarity of genome-wide patterns of enhancer chromatin accessibility.

We found that whole retina DNase-seq clustered with bipolar cell samples (*Figure 3A*). This finding was unexpected given that rod photoreceptors constitute ~ 80% of all cells in the mouse retina (*Figure 1A*) (*Jeon et al., 1998*) and indicates that the open chromatin profile of a complex tissue is not necessarily dominated by the most abundant cell type. This is in line with our previous comparison of genome-wide patterns of chromatin accessibility in rods, cones and whole retina, which showed that more than half of the whole retina peaks did not overlap with photoreceptor peaks, suggesting that they derive from non-photoreceptor retinal cell types (*Hughes et al., 2017*). This finding may be a consequence of the distinctive pattern of global chromatin closure that we previously described in rod photoreceptors (*Hughes et al., 2017*).

We next set out to examine patterns of chromatin accessibility within the retinal cell types in greater detail. We combined TSS-distal ATAC-seq peaks from bipolar cells and photoreceptors with DNase-seq peaks from whole retina to create a list of 99,684 retinal open chromatin regions. Clustering these regions based on chromatin accessibility across retinal and non-retinal cell types offers a broad view of cell class- and cell type-specific regions of open chromatin (*Figure 3B*). We found a large subset of bipolar-enriched peaks, many of which were also accessible in whole retina and brain (*Figure 3B*, green box). Conversely, a smaller subset of peaks showed selective accessibility in photoreceptors, with lower levels of accessibility in bipolar cells, and even less in other cell types (*Figure 3B*, red box). While pan BCs and ON-BCs showed nearly identical open chromatin profiles, OFF-BC open chromatin patterns were somewhat divergent, with slightly more accessibility in the photoreceptor-enriched subset (red box) and less in the bipolar-enriched subset (green box) compared to ON-BC. Overall, the sum of photoreceptor and bipolar cell ATAC-seq peaks accounted for 83 percent of whole-retina DNase-seq peaks, with the remainder presumably deriving from other inner retinal cell classes. These data suggest that relatively few regions of open chromatin are truly photoreceptor-specific, and that regions enriched in bipolar cells are more likely to be accessible in other tissues, particularly in brain. For instance, compared to photoreceptors, a greater proportion of bipolar cell peaks corresponded to DNase-seq peaks in brain (47% vs 37%). Finally, direct comparison of ATAC-seq peaks from photoreceptor and bipolar cells identified 55,402 differentially accessible regions (FDR < 0.05), 75% of which are more accessible in bipolar cells (*Supplementary file 7*).

While we observed significant differences in global chromatin accessibility between photoreceptors and bipolar cells, the open chromatin profiles of ON and OFF bipolar cells were largely similar,

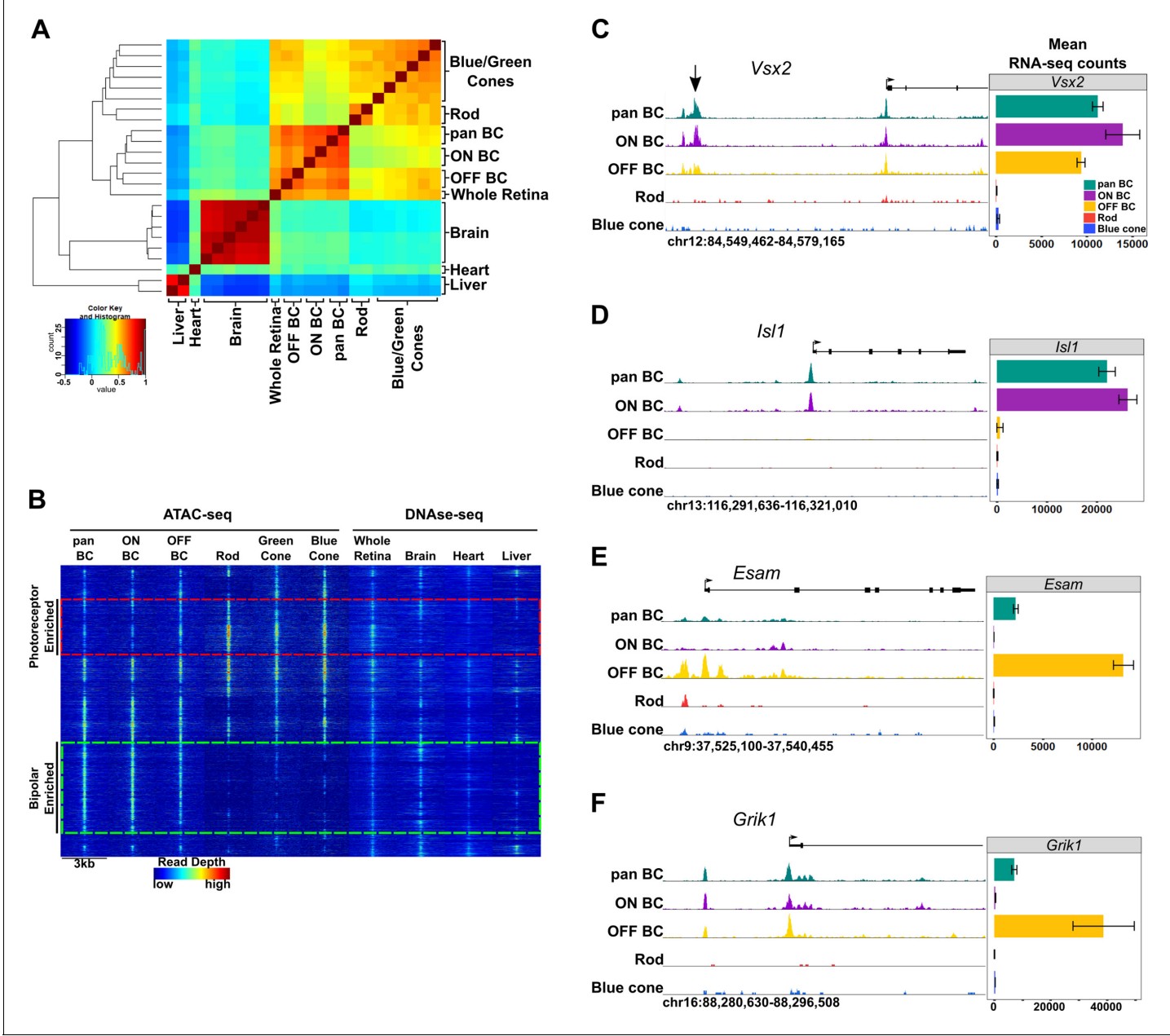

**Figure 3.** Genome-wide open chromatin profiles of ON and OFF bipolar cells. (A) Heatmap showing pairwise correlation between ATAC-seq data from photoreceptor and bipolar cell populations as well as DNase-seq data from whole retina, brain, heart and liver. Multiple biological replicates are shown for most tissues. Peaks from each sample were combined to generate a set of 302,518 enhancer peaks, and replicates were clustered based on read counts at each peak. Bipolar cells and photoreceptors form separate clusters. Whole retina DNase-seq clusters with bipolar cells. Photoreceptors, bipolar cells and whole retina cluster separately from other tissues. (B) Genome-wide profiles of chromatin accessibility in isolated photoreceptor and bipolar cell ATAC-seq datasets as well as DNase-seq datasets from additional control tissues. Rows show accessibility as indicated by read depth in 3 kb windows centered on peak summits sampled from photoreceptor, bipolar, and whole retina datasets (10,000 peaks randomly sampled from a total of 99,684 enhancer peaks are shown). Hierarchical clustering reveals peak sets enriched in photoreceptors (red box) or bipolar cells (green box). (C–F) Screenshots of UCSC genome browser tracks show regions of accessible chromatin in bipolar and photoreceptor populations at loci that exhibit shared or cell class-specific expression patterns. Black arrow in panel C indicates a known enhancer of *Vsx2* (***Kim et al., 2008***). There is an imperfect correlation between chromatin accessibility and gene expression. Bar graphs aligned with browser tracks indicate mean RNA-seq counts of each gene for the indicated populations.

DOI: https://doi.org/10.7554/eLife.48216.009

consistent with the high degree of similarity between their transcriptomes. Specifically, only 4263 peaks were differentially accessible between ON and OFF bipolar cells. Of note, 79% (3359) of these differential peaks were more accessible in ON bipolar cells. When examining loci surrounding genes expressed in both ON and OFF bipolar cells, we found that ON and OFF subclasses typically had similar open chromatin profiles (e.g., *Vsx2*, *Figure 3C*). In contrast, some ON- or OFF-specific genes exhibit cell subclass-specific patterns of chromatin accessibility (e.g., *Isl1* and *Esam*, *Figure 3D–E*). We found an overall correlation between accessibility and associated gene expression (*Figure 5— figure supplement 1*) but did not see this pattern at all gene loci (e.g., *Grik1*, *Figure 3F*).

## Photoreceptor and bipolar cells employ closely related yet distinct *cis*-regulatory grammars

Having compared global accessibility between photoreceptor and bipolar cells, we next sought to compare them in terms of '*cis*-regulatory grammar', which we define as the number, affinity, spacing and orientation of TF binding sites within the open chromatin regions of a given cell type or class. To begin, we assessed all 319 TF binding site motifs from the HOMER database for enrichment within bipolar cell open chromatin regions (*Heinz et al., 2010*). The most highly enriched motifs within enhancers corresponded to CTCF, K50 HD, E-box, nuclear receptor, and MADS box motifs (*Supplementary file 8*). All of these motifs were previously shown to be among the most enriched motifs in photoreceptor ATAC-seq peaks as well (*Hughes et al., 2017*). The similarity in the patterns of TF binding site enrichment between photoreceptors and bipolar cells can be better understood in the context of known patterns of TF expression in these cell classes. Specifically, the K50 HD TFs OTX2 and CRX are master regulators of gene expression programs in photoreceptor and bipolar cells. Likewise, bHLH TFs (which recognize E-box motifs) play roles in fate specification and maintenance of both photoreceptor and bipolar cell gene expression programs, as described in the Introduction. Finally, enrichment of ZF motifs recognized by CTCF in both cell classes is in line with reports that CTCF motifs often lie in ubiquitously accessible chromatin regions, where CTCF recruitment is thought to mediate interactions between promoters and enhancers, among other functions (*Splinter et al., 2006*; *Rao et al., 2014*; *Song et al., 2011*).

Despite the overall similarity between the *cis*-regulatory grammars of bipolar cells and photoreceptors, there are notable quantitative differences in motif enrichment between the two cell classes. To systematically identify these differences, we compared the proportion of peaks containing each of the 319 motifs between pan-bipolar cells and each photoreceptor cell type (*Supplementary file 8*). The most differentially enriched motifs corresponded to those with the highest enrichment in each cell class and are summarized in *Figure 4*. Although both photoreceptors and bipolar cells showed marked enrichment for K50 HD motifs, these motifs were more enriched in photoreceptors (*Figure 4*). Conversely, E-box motifs were more enriched in bipolar cells than photoreceptors. The most striking difference in TF binding site enrichment between the two cell classes was the enrichment of both monomeric and dimeric Q50 HD motifs in photoreceptor open chromatin regions and their lack of enrichment in bipolar regions (*Figure 4*). The most well-characterized Q50 HD TF expressed in photoreceptors is RAX, which is required for cone gene expression and survival (*Irie et al., 2015*). In contrast, bipolar cells express multiple Q50 HD TFs (VSX2, VSX1, ISL1, LXH3, LHX4, AND SEBOX) (*Shekhar et al., 2016*). VSX2 is required for bipolar cell development and is expressed in all mouse bipolar cell types throughout development and into adulthood (*Livne-Bar et al., 2006*; *Liu et al., 1994*). The paradoxical absence of Q50 HD motif enrichment in bipolar open chromatin regions despite the presence of multiple Q50 HD TFs in this cell class may be explained by the observation that VSX2 acts as a repressor of photoreceptor CREs (*Dorval et al., 2006*). The lack of Q50 motif enrichment in bipolar cells could be due to selective repression of photoreceptor-specific open chromatin regions by VSX2, which, in turn, prevents ectopic expression of photoreceptor genes in bipolar cells, a possibility that we will return to in the final section of the Results.

To further compare the *cis*-regulatory grammars of bipolar cells and photoreceptors, we examined TF binding site co-occurrence and spacing within each cell class. For this analysis we compared a combined list of enhancer regions from rod and cones to that of bipolar cells. As expected, motif pairs enriched in specific peak sets tended to include motifs that showed the highest individual enrichment in the same cell type. Likewise, differentially enriched motif pairs included one or more differentially enriched motifs, such as K50 and Q50 HD motifs in photoreceptors and bHLH motifs in

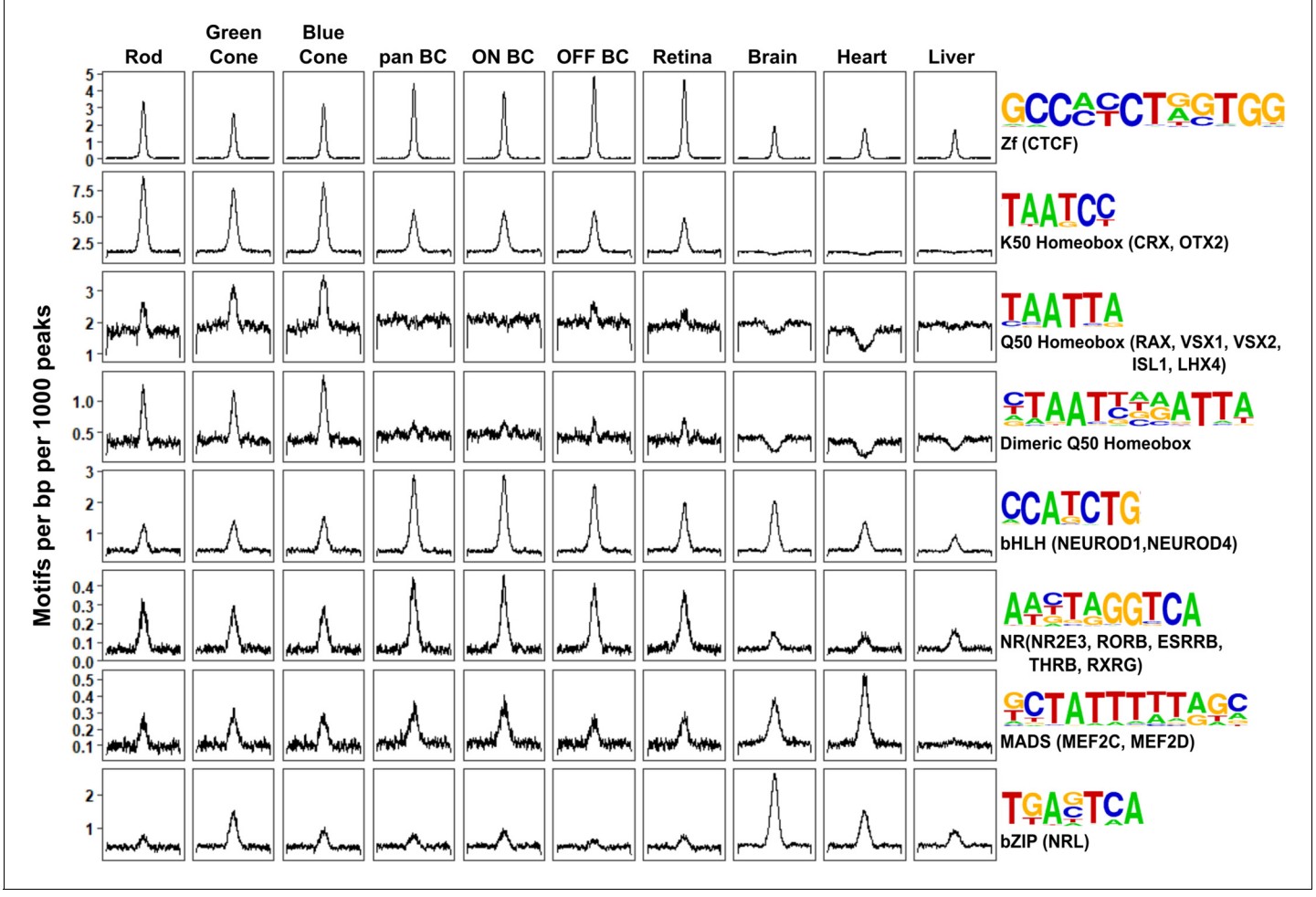

**Figure 4.** Patterns of TF binding site enrichment across retinal cell classes. Motif enrichment patterns identified in bipolar and photoreceptor ATAC-seq datasets as well as in DNase-seq datasets from adult mouse whole retina, brain, heart, and liver. This analysis included all enhancer (i.e., TSS-distal) open chromatin peaks from each cell class or tissue. Each panel is centered on a 1 kb window around peaks from the indicated dataset. Motif density (motifs per base pair per 1000 peaks) is shown on the Y-axis. Consensus sequences for each motif class and example TFs (in parentheses) expressed in photoreceptors and bipolar cells are shown on the right. Photoreceptor and bipolar cell populations share enrichment for K50 HD motifs, while only photoreceptors show enrichment for Q50 HD motifs.

DOI: https://doi.org/10.7554/eLife.48216.010

The following figure supplements are available for figure 4:

**Figure supplement 1.** Enrichment of motif pairs in photoreceptors and bipolar cells.

DOI: https://doi.org/10.7554/eLife.48216.011

**Figure supplement 2.** Motif spacing and orientation in photoreceptor and bipolar cells.

DOI: https://doi.org/10.7554/eLife.48216.012

bipolar cells (*Figure 4—figure supplement 1A–B*). Of note, while we identified specific motif pairs enriched in each cell type, the frequencies of these pairs are consistent with an independent model (i.e., the number of occurrences of motif pairs is approximately what would be expected given the number of occurrences of each individual motif and assuming a random distribution of motifs across peaks). As with chromatin accessibility, enrichment of motif pairs is highly similar between ON and OFF bipolar cells, with nominally differentially enriched pairs showing similar proportions (*Figure 4— figure supplement 1C*). Finally, to investigate preferences in spacing and orientation between pairs of motifs, we plotted the density of highly enriched motifs (those depicted in *Figure 4*) centered on regions flanking K50 and Q50 HD motifs in each peak set. As described previously for photoreceptors, spacing and orientation preferences in bipolar open chromatin regions were minimal (*Figure 4—figure supplement 2*) (*Hughes et al., 2017*). Thus, the primary differences in the *cis-*

regulatory grammar of the two cell classes appears to be the degree of HD and E-box motif enrichment.

## Photoreceptor- and bipolar-specific open chromatin regions are positively correlated with cell class-specific gene expression

We next sought to determine the extent to which photoreceptor- and bipolar-enriched open chromatin regions correlate with cell type-specific gene expression. To this end, we assigned each of the 55,402 regions identified as differentially accessible between photoreceptor and bipolar cells to a candidate target gene based on proximity to the nearest transcription start site and compared mean RNA-seq expression values for the assigned genes. As described in previous studies, we observed a modest correlation between enhancer accessibility and gene expression, and a more robust correlation between promoter accessibility and gene expression (*Figure 5—figure supplement 1*) (*Hughes et al., 2017*; *Pastor et al., 2014*; *Ampuja et al., 2017*; *de la Torre-Ubieta et al., 2018*).

Our analysis of global chromatin accessibility suggested that many of the differentially accessible peaks were also open in other tissues, especially those enriched in bipolar cells compared to photoreceptors (*Figure 3E*). To gain a better understanding of the cell type-specific open chromatin regions that drive gene expression differences between these two cell classes, we refined our analysis to exclude peaks shared with non-retinal cell types. We identified 8435 enhancer regions which are accessible either in photoreceptors or bipolar cells, but not accessible in brain, liver or splenic B cells (*Figure 5A*). This set includes 1291 regions that are open in both photoreceptors and bipolar cells, and 7144 regions that are differentially accessible between the two cell classes (*Figure 5A*). We found that ∼ 46% (3,270) of the differentially accessible peaks were more open in bipolar cells. Thus, most of the bipolar cell-enriched regions identified in the previous section were also accessible in one or more non-retinal tissues. As with the enhancer regions from the unfiltered list, assigning genes to this more retina-specific set of differentially accessible regions also shows a correlation between accessibility and gene expression (*Figure 5B*). To visualize this association and identify the peaks that underly it, we plotted all 8435 peaks according to fold-change differences in accessibility and gene expression between bipolar cells and rods (*Figure 5C*) and between bipolar cells and blue cones (*Figure 5—figure supplement 1D*). We then selected for further analysis those peaks that exhibited correlated accessibility and gene expression in photoreceptors (highlighted in red or blue in *Figure 5C* and *Figure 5—figure supplement 1D*; n = 901) or bipolar cells (highlighted in green in *Figure 5C* and *Figure 5—figure supplement 1D*; n = 833). These differentially enriched peaks represent strong candidates for CREs that mediate the gene expression differences between the two cell classes. Indeed, the photoreceptor peak set contains known enhancers responsible for driving cell type-specific expression of *Rhodopsin* (*Rho*) and components of the rod-specific phototransduction cascade (*Corbo et al., 2010*; *Nie et al., 1996*), while the bipolar peak set contains a known enhancer that drives *Vsx2* expression in bipolar cells (*Kim et al., 2008*). To gain insight into the possible biological functions of these peaks we used GREAT to assign biological annotations based on nearby genes (*McLean et al., 2010*). As was found with the unfiltered datasets, photoreceptor peaks are linked with genes associated with light sensation, whereas bipolar peaks are linked to genes involved in more generic neuronal functions (*Figure 5—figure supplement 2*).

Next, we asked whether the patterns of TF binding site enrichment observed with aggregate sets of ATAC-seq peaks from each cell class would be preserved within the retina-specific peak sets associated with correlated gene expression. We compared photoreceptor-enriched regions, bipolar-enriched regions, and regions that share accessibility between the two cell classes which were either specific to the retina (*Figure 5A*, 'shared retina' n = 1,291), or unfiltered ('shared all', n = 47,550). We found that K50 HD motifs were enriched in both shared and cell class-selective regions, but to a lesser extent in regions specifically enriched in bipolar cells. The bipolar-selective regions were markedly enriched for E-box motifs but completely lacked enrichment for Q50 HD motifs (*Figure 5D*). Conversely, photoreceptor-selective regions were enriched for Q50 HD motifs, but lacked E-box motif enrichment. Peaks that were shared between the two cell classes showed an intermediate pattern of motif enrichment, highlighting the roles of both HD and bHLH TFs in regulating the gene expression programs of each class. Of note, CTCF enrichment was absent in all but the unfiltered peak set, suggesting that the CTCF enrichment observed in *Figure 4* is attributable to ubiquitously accessible peaks. Taken together, these findings suggest that differential enrichment of

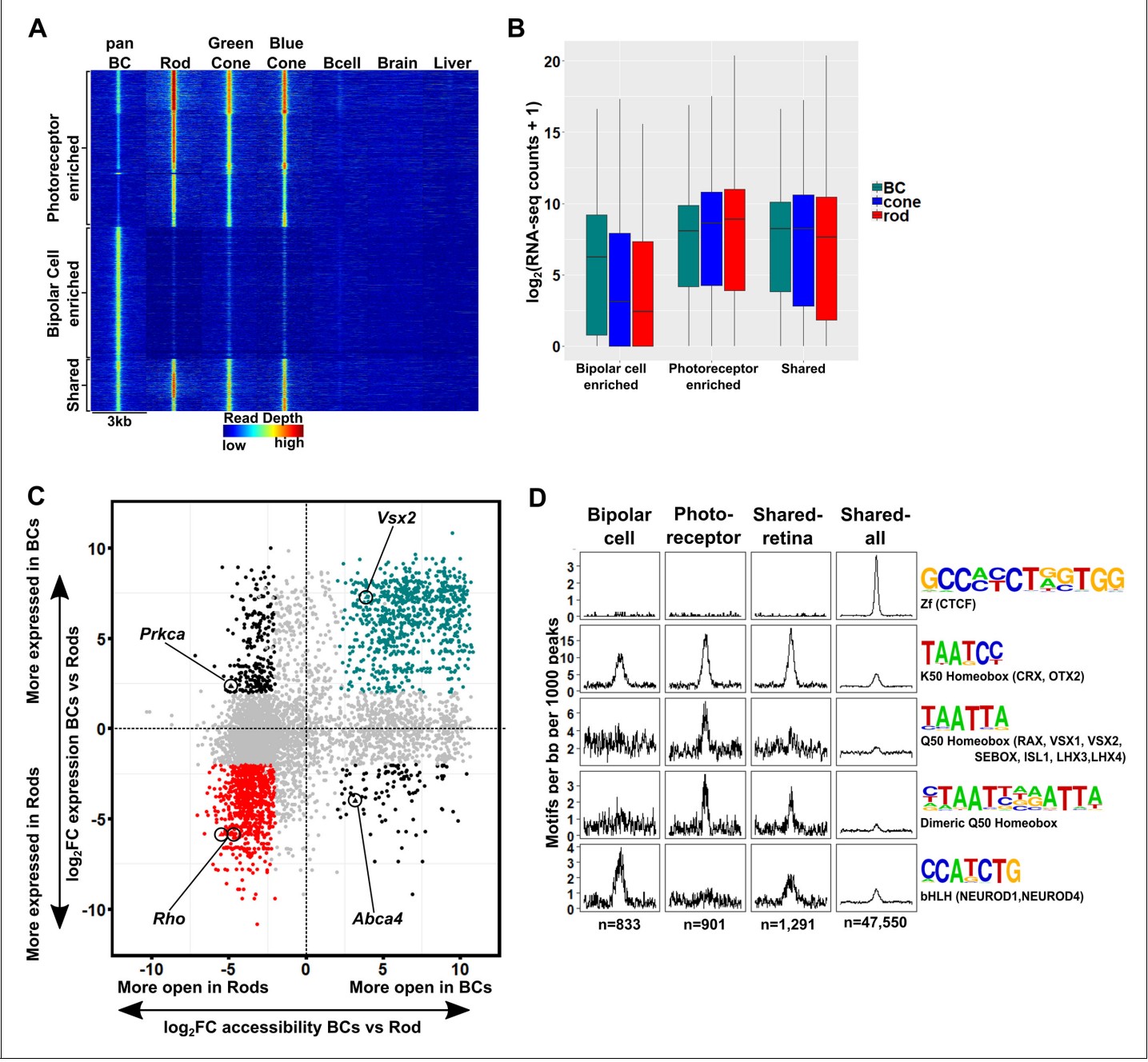

**Figure 5.** The *cis*-regulatory grammar of genomic regions associated with differential chromatin accessibility and gene expression in bipolar cells and photoreceptors. (**A**) Heatmap depicting 8,435 genomic regions determined by pairwise comparison to be differentially accessible in photoreceptors, bipolar cells, or both cell classes, compared to adult mouse B cells, brain and liver. Photoreceptor-enriched (n = 3,874), bipolar cell-enriched (n = 3,270), shared (n = 1,291). (**B**) Expression of genes to which classes of peaks defined in A were assigned by proximity to nearest TSS. There is a moderate correlation between chromatin accessibility and gene expression in each cell class. (**C**) Peaks identified in panel A are plotted according to chromatin accessibility (x-axis) and associated gene expression (y-axis) in bipolar cells versus rods. Peaks with four-fold greater chromatin accessibility and associated gene expression in bipolar cells are shown in green (FDR < 0.05 for both accessibility and expression), while those peaks with greater accessibility and associated gene expression in rods are shown in red. Shared peaks and those associated with modest (less than four-fold) differences in expression are in gray. Peaks with discordant chromatin accessibility and associated gene expression are shown in black. Peaks assigned to genes expressed specifically in photoreceptors (*Rho*, *Abca4*) or bipolar cells (*Vsx2*, *Prkca*) are indicated. (**D**) Motif enrichment within peaks displaying correlated chromatin accessibility and associated gene expression in bipolar cells and photoreceptors as well as within peaks displaying shared accessibility, including those shown in panel A (shared-retina) and shared peaks which were not filtered to remove those accessible in non-retinal tissues (shared-all). Both photoreceptor and bipolar cell peaks show enrichment for K50 HD motifs. Bipolar cell peaks (n = 833) are also highly enriched

*Figure 5 continued on next page*

*Figure 5 continued*

for E-box motifs but lack enrichment of Q50 HD motifs. In contrast, photoreceptor peaks (n = 901) show enrichment for Q50 HD motifs but almost entirely lack E-box enrichment. Shared retina-specific peaks (n = 1,291) show a hybrid pattern of motif enrichment. Only the shared-all peak set exhibits enrichment for CTCF motifs, underscoring a key difference between cell-class specific open chromatin regions (which show no enrichment of CTCF motif) and ubiquitously open chromatin regions, which show strong enrichment.

DOI: https://doi.org/10.7554/eLife.48216.013

The following figure supplements are available for figure 5:

**Figure supplement 1.** Correlation between chromatin accessibility and expression of associated genes.
DOI: https://doi.org/10.7554/eLife.48216.014

**Figure supplement 2.** Gene Ontology (GO) annotation of peaks exhibiting correlated chromatin accessibility and associated gene expression.
DOI: https://doi.org/10.7554/eLife.48216.015

Q50 HD and E-box motifs are the key features that distinguish the *cis*-regulatory grammars of photoreceptors and bipolar cells.

## K50 motifs in the *Gnb3* promoter are required for both photoreceptor and bipolar expression, but addition of Q50 motifs selectively represses expression in bipolar cells

Given the critical roles of HD TFs in the regulation of photoreceptor and bipolar gene expression, we further investigated the role of K50 and Q50 motifs in the *cis*-regulatory region upstream of *Gnb3*. *Gnb3* encodes the β subunit of a heterotrimeric G-protein required for cone phototransduction as well as ON bipolar cell function (*Dhingra et al., 2012*). *Gnb3* is expressed in rods, cones, and bipolar cells during early postnatal retinal development in the mouse. Selective repression of *Gnb3* in rods by the nuclear receptor TF NR2E3 results in a cone + bipolar pattern after postnatal day 10 (*Haider et al., 2009*; *Corbo and Cepko, 2005*). We focused on an 820 bp region around the TSS of *Gnb3* which drives robust expression in rods, cones, and bipolar cells when electroporated into early postnatal mouse retina. This region lacks Q50 motifs but contains five K50 HD motifs of varying affinity which occur in two clusters, one immediately upstream of the TSS (−65 bp) and the other more distally (−350 bp) (*Figure 6A*). To evaluate the role of these five K50 motifs in mediating photoreceptor and bipolar expression, we engineered reporter constructs in which each of the five motifs was individually inactivated by mutating the TAAT core to TGGT. We then introduced wild-type and mutant reporters into mouse retinal explants via electroporation and compared expression levels after 8 days. Mutations in K50 motifs 2, 4 and 5 resulted in coordinate loss of expression in both photoreceptor and bipolar cells, indicating that these motifs are required for reporter expression in both cell classes (*Figure 6B,D*). Conversely, mutations in site 1 or 3 had no effect on expression in either cell class (*Figure 6B,D*). Binding site affinity did not correlate with expression, as site 3 has a higher predicted affinity than sites 4 or 5. Thus, the *Gnb3* promoter region contains both essential and nonessential K50 motifs, underscoring the critical role for these shared motifs in both photoreceptors and bipolar cells.

Next, we sought to determine the effect of introducing Q50 motifs into the *Gnb3* promoter region. For these experiments, reporters were introduced into newborn mouse retinas via in vivo electroporation and harvested for histologic analysis after 20 days. First, we electroporated identical wild-type sequences driving both DsRed and GFP to confirm that essentially all photoreceptors and bipolar cells received both constructs (*Figure 6C*). Next, we compared the expression of a *Gnb3* promoter containing mutations in K50 motifs 1 and 3 to that of a wild-type promoter, confirming that elimination of both of these sites has no effect on expression in either photoreceptors or bipolar cells (*Figure 6C,D* and *Figure 6—source data 1*). To test the effect of introducing Q50 motifs into the *Gnb3* promoter region, we replaced K50 motifs 1 and 3 with a Q50 motif (TAATTA), both individually and in combination. Whereas introduction of a Q50 motif into site one had no apparent effect, replacement of site three caused a selective decrease in bipolar expression with no change in photoreceptor expression. When we introduced Q50 motifs into both sites, reporter expression was markedly reduced in bipolar cells, with no effect on photoreceptor expression (*Figure 6C,D* and *Figure 6—source data 1*). These data, along with previous reports of VSX2-mediated repression of photoreceptor-specific promoters and enhancers, suggest that Q50 motifs play an important role in mediating repression of photoreceptor genes in bipolar cells.

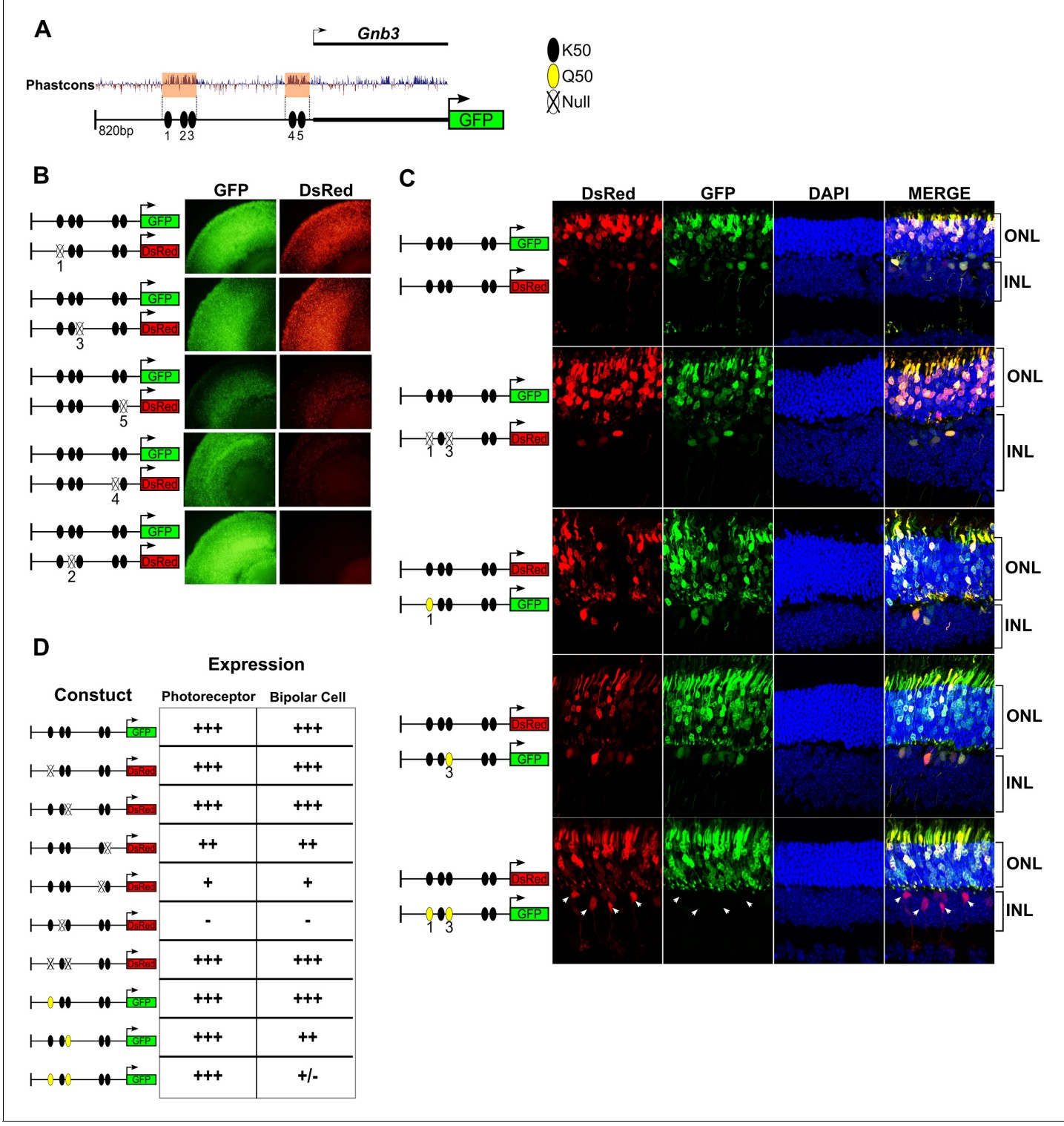

**Figure 6.** K50 motifs are required for expression in both photoreceptors and bipolar cells, while Q50 motifs mediate repression of reporter expression specifically in bipolar cells. (**A**) Schematic of reporter construct containing 820 bp from the promoter region and 5' untranslated region (UTR) of mouse *Gnb3*. This region contains two phylogenetically conserved blocks harboring a total of five K50 motifs (black ellipses). (**B**) Left, Schematics of reporter pairs containing wild-type (WT; black) and inactivated (crossed out) K50 motifs. Right, Each pair of reporters was electroporated into explanted newborn mouse retina, which were subsequently harvested after eight days in culture and photographed in a flat mount preparation. While K50 sites 2, 4 and 5 are required for reporter expression, sites 1 and 3 are dispensable. (**C**) Left, Pairs of reporters containing WT, inactivated K50, or novel Q50 (TAATTA; yellow) motifs were electroporated into newborn mouse retina in vivo. After 20 days, the retinas of electroporated mice were harvested and

*Figure 6 continued on next page*

*Figure 6 continued*

photographed in vertical cross-sections. (**C**) Right, Representative cross-sections of retinas injected with the indicated pair of reporters. Loss of both K50 motifs 1 and 3 has no effect on expression in either photoreceptors or bipolar cells, while converting these same sites to Q50 motifs abrogates expression specifically in bipolar cells (n = 2–7 depending on reporter pair, ***Figure 6—source data 1***). White arrowheads indicate bipolar cells expressing the WT reporter, but not the mutated one. (**D**) Table summarizing the results of reporter analysis presented in (**B**) and (**C**).

DOI: https://doi.org/10.7554/eLife.48216.016

The following source data is available for figure 6:

**Source data 1.** Additional images of *Gnb3* reporter injections.
DOI: https://doi.org/10.7554/eLife.48216.017

## Discussion

In this study we generated open chromatin maps and transcriptome profiles of mouse bipolar cells, including FACS-purified ON and OFF bipolar cell populations, and compared them to analogous data from rod and cone photoreceptors. We found that photoreceptors and bipolar cells differ in the expression of thousands of genes and yet have very similar *cis*-regulatory grammars. The key *cis*-regulatory differences that distinguish the two cell classes are the preferential enrichment of Q50 HD motifs in open chromatin regions associated with photoreceptor-specific gene expression and a corresponding enrichment of E-box motifs in chromatin associated with bipolar-specific expression. The cellular features and transcriptional mechanisms shared by photoreceptors and bipolar cells have prompted speculation that these two sister cell types arose from a single ancestral photoreceptor cell type via a process of progressive cellular divergence (***Arendt, 2008***; ***Lamb, 2013***). We propose that the elimination of Q50 motifs from bipolar-specific CREs likely played a key role in differentiating the bipolar transcriptome from that of photoreceptors during early stages of vertebrate retinal evolution. Alternatively, given the multiplicity of K50 HD binding sites within both photoreceptor and bipolar cell regulatory elements, a subset of photoreceptor-specific elements may have emerged through the simple conversion of K50 (TAATCC) to Q50 (TAATT$^A$/$_G$) sites. Prior studies of individual photoreceptor CREs showed a role for the Q50 HD TF, VSX2, in selectively repressing photoreceptor genes in bipolar cells (***Livne-Bar et al., 2006***; ***Dorval et al., 2006***). Our results generalize this conclusion, suggesting that VSX2 plays a genome-wide role in silencing photoreceptor gene expression in bipolar cells. A similar role for VSX2 has recently been described in the spinal cord, where closely related progenitor cells give rise to either motor neurons or V2a interneurons (***Clovis et al., 2016***). VSX2 promotes V2a identity by directly repressing the motor neuron gene expression program and by competing for Q50 sites at motor neuron enhancers. Thus, in both retina and spinal cord, expression of VSX2 promotes interneuron fate at the expense of the alternative neuronal (photoreceptor or motor neuron) cell type. These parallels suggest that transcriptional repression by cell type-specific TFs such as VSX2 represent a common mechanism for differentiating the gene expression programs of two closely related cell types.

In addition to differential enrichment of Q50 sites, we also observed enrichment of E-box motifs in regions associated with bipolar-specific expression (***Figure 5C***). The lack of corresponding enrichment in regions associated with photoreceptor-specific expression suggests that bHLH TFs also play an important role in distinguishing the gene expression programs of photoreceptor and bipolar cells. It is important to note that this finding does not contradict known roles for bHLH TFs in photoreceptors, as E-box motifs are also enriched within the complete set of photoreceptor ATAC-seq peaks (***Figure 4***), as well as in the subset that exhibits shared accessibility with bipolar cells (***Figure 5C***). The role of bHLH TFs in establishing the cellular identity of both classes is further demonstrated by multiple loss-of-function studies (***Tomita et al., 2000***; ***Bramblett et al., 2004***; ***Feng et al., 2006***; ***Huang et al., 2014***; ***Pennesi et al., 2003***).

In contrast to the differences between photoreceptors and bipolar cells, ON and OFF bipolar cell populations displayed striking similarities in their *cis*-regulatory landscape and gene expression profiles. In a paper that appeared after our profiling studies had been performed, ***Shekhar et al. (2016)*** documented the single-cell expression profiles of individual bipolar cell types and showed that the transcriptomes of ON and OFF cone bipolar cells are more similar to each other than to that of rod bipolar cells (RBCs). This finding implies that the ON bipolar population analyzed in the present study represents a grouping of distinct bipolar cell types (RBC and cone ON BC), which

should be separated for a more informative comparison. Despite this drawback, our analysis indicates that there are relatively few differences in chromatin accessibility between bipolar cell types. We estimate that RBCs constitute nearly 60% of the cells in our ON population (RBCs compose 56% of ON bipolar cells identified in *Shekhar et al., 2016*, and ~ 57% of those identified as ON BCs by *Wässle et al., 2009*). Thus, much of the signal in our ON bipolar RNA-seq and ATAC-seq data likely derives from RBCs. It remains to be determined whether open chromatin profiling of individual bipolar subtypes will reveal additional differences in their epigenomic landscapes beyond those reported here.

We found an imperfect correlation between chromatin accessibility and gene expression in bipolar cells. For example, ATAC-seq peaks upstream of *Grik1* are open in ON bipolar cells despite an absence of *Grik1* expression in this cell type (*Figure 3F*). We observed similar instances of discordance between chromatin accessibility and gene expression in our previous analysis of rod and cone photoreceptors (*Hughes et al., 2017*), and other groups have documented this discrepancy in other cell types (*de la Torre-Ubieta et al., 2018*; *Lara-Astiaso et al., 2014*; *Starks et al., 2019*). Presumably, transcriptional activity in these instances requires expression of additional cell type-specific factors (*Heinz et al., 2015*), perhaps most clearly demonstrated in developmental contexts, wherein accessibility is frequently established prior to the onset of transciption (*Lara-Astiaso et al., 2014*).

Support for the idea that bipolar cells diverged from photoreceptors via progressive partitioning of cellular function is provided by the existence of cell types in the retinas of non-mammalian vertebrates with features intermediate between those of mammalian photoreceptors and bipolar cells. In some turtle species ~ 30% of the cell bodies in the photoreceptor layer (the outer nuclear layer) belong to bipolar cells, not photoreceptors (*Tauchi, 1990*). These so-called 'displaced bipolar cells' possess an inner segment-like process that extends to the outer limiting membrane and contains abundant mitochondria and even a sensory-type ('9 + 0') cilium (*Figure 7A,B*). Thus, displaced bipolar cells closely resemble typical photoreceptors except that they lack an outer segment, possess dendrites in the outer plexiform layer, and synapse directly onto retinal ganglion cells. Another intermediate type of bipolar cell occurs in nearly all non-mammalian vertebrate classes and even in some mammalian species (*Hendrickson, 1966*; *Locket, 1970*; *Quesada and Génis-Gálvez, 1985*; *Young and Vaney, 1990*). This bipolar type has a nucleus localized to the inner nuclear layer, but retains an inner segment-like structure (Landolt's club), which extends from the cell's dendritic arbor to the outer limiting membrane and contains abundant mitochondria and a sensory-type cilium (*Figure 7A*) (*Hendrickson, 1966*; *Locket, 1970*; *Quesada and Génis-Gálvez, 1985*). We suggest that displaced bipolar cells and those with a Landolt's club represent 'transitional forms' on the evolutionary path from photoreceptor to typical bipolar cell. The existence of these transitional forms suggests that bipolar cells may have evolved via the stepwise repression of discrete gene modules required for the development of individual cellular features, or 'apomeres', that are specific to photoreceptors (*Arendt et al., 2016*).

If this evolutionary model is correct, then how can we account for the co-existence of 'transitional' bipolar cell types and 'conventional' bipolar cells in a single retina? One testable hypothesis is that VSX2 may be expressed at lower levels in transitional bipolar cell types, thereby permitting expression of additional photoreceptor gene modules and their corresponding apomeres. Alternatively, it is possible that additional activating Q50 HD TFs are expressed in transitional bipolar types, and these TFs can overcome VSX2-mediated repression of selected photoreceptor gene modules. Indeed, we and others have found that multiple Q50 HD TFs are expressed in subsets of mouse bipolar cells (*Shekhar et al., 2016*). In addition, there is evidence that transitional bipolar cell types with Landolt's club may exist in the mouse (*Rowan and Cepko, 2004*). Thus, individual bipolar cell types may control the number of photoreceptor apomeres they express by modulating the balance of activating and repressing Q50 HD TFs in their nuclei.

The evolutionary divergence of bipolar cells from photoreceptors likely required coordinated changes in both *cis*-regulatory grammar and HD TF expression. The Q50 HD TF, RAX, is expressed in developing vertebrate rods and cones and is required for normal activation of photoreceptor gene expression in mice (*Irie et al., 2015*; *Rodgers et al., 2018*). The expression of a RAX homolog in the photoreceptors of the tadpole larva of the protochordate, *Ciona intestinalis*, suggests a primordial role for this Q50 HD TF in activating photoreceptor gene expression in chordates (*Cao et al., 2019*). These data suggest that both K50 and Q50 motifs were present in the CREs of the ancestral vertebrate photoreceptor prior to the evolutionary emergence of bipolar cells, and

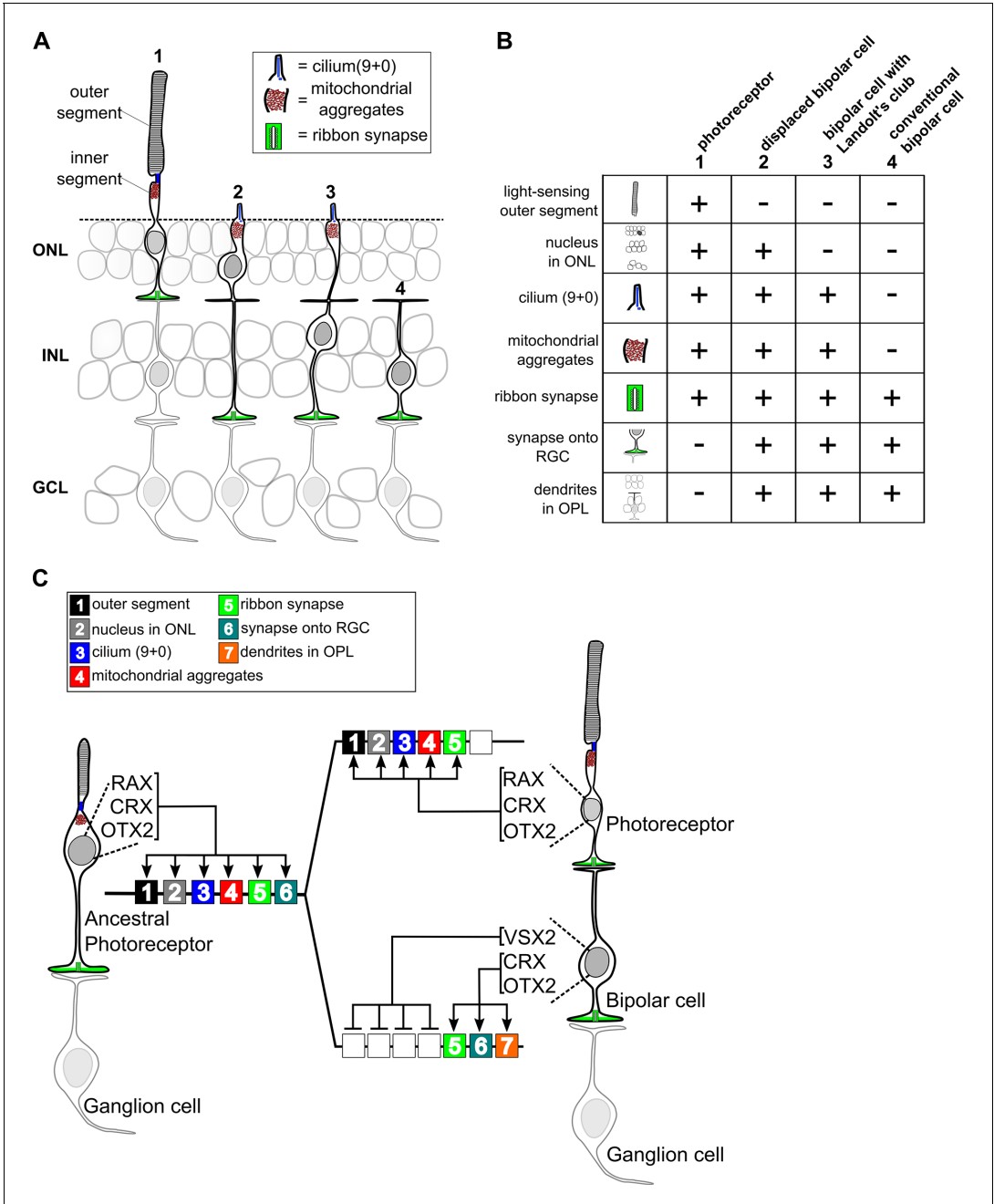

**Figure 7.** Evolutionary model for the divergence of bipolar cells from photoreceptors. (A) Schematic depiction of photoreceptors (1), conventional bipolar cells (4), and two 'transitional' cell types with cellular features intermediate between those of photoreceptors and conventional bipolar cells: displaced bipolar cells (2) and bipolar cells with Landolt's club (3). (B) Table of individual cellular features (referred to in Arendt et al. as 'apomeres') possessed by photoreceptors, bipolar cells, or both cell classes. (C) Evolutionary model for the divergence of present-day photoreceptors and bipolar cells from a common ancestral photoreceptor type. We propose that the ancestral photoreceptor (possibly present in a hagfish-like ancestor) expressed cell type-specific genes via both K50 HD TFs (CRX and OTX2) and a possibly weakly activating Q50 HD TF (RAX). The emergent expression of a strongly repressive Q50 HD TF (VSX2) in bipolar cells then permitted silencing of selected photoreceptor gene modules underlying the formation of defined photoreceptor apomeres (e.g., outer segment). Selective expression of activating Q50 HD TFs in 'transitional' bipolar cell types may have allowed the derepression of specific photoreceptor apomeres (e.g., cilium formation, mitochondrial aggregates). Novel bipolar-specific apomeres (e.g., dendrites in the outer plexiform layer [OPL]) may have evolved via co-option of other gene expression programs.
DOI: https://doi.org/10.7554/eLife.48216.018

that both K50 (OTX2 and CRX) and Q50 (RAX) HD TFs were required for gene activation in that ancestral cell type (*Figure 7C*). In this context, the emergent expression of a repressive Q50 HD TF (VSX2) in a primordial bipolar cell would have permitted selective repression of CREs containing Q50 motifs, allowing the *cis*-regulatory landscape of the two nascent cell types to begin to diverge. Maintaining expression of selected 'photoreceptor' genes in bipolar cells (e.g., *Gnb3*) would then have required the elimination of Q50 motifs from the *cis*-regulatory regions of those genes.

These evolutionary considerations suggest that the modern vertebrate retina arose from an ancestral retina in which photoreceptors directly synapse onto projection neurons (i.e., ganglion cells) without an intervening layer of interneurons (left side of *Figure 7C*). Two lines of evidence suggest that such a retina may have existed. First, the retina of the hagfish, the most primitive extant vertebrate, reportedly has photoreceptors that directly synapse onto projection neurons (i.e., ganglion cells) (*Lamb, 2013*; *Locket and Jørgensen, 1998*). Second, some vertebrate species (including reptiles, amphibians, and larval lamprey) have an unpaired, median 'parietal eye' developmentally related to the pineal gland, which contains photoreceptors that directly synapse onto ganglion cells (*Eakin, 1973*; *Dodt, 1973*; *Mano and Fukada, 2007*; *Solessio and Engbretson, 1993*). It is possible that the parietal eye evolved from the midline 'eye' of a protochordate ancestor, akin to the present-day ascidian larva. The simple lateral eyes of a hagfish-like vertebrate ancestor may then have emerged via co-option of the gene networks required for parietal eye development. Subsequently, bipolar cells may have arisen in the lateral eyes of early vertebrates via subtle changes in *cis*-regulatory grammar and TF expression, paving the way for the emergence of the sophisticated interneuronal circuitry found in present-day vertebrate retinas.

# Materials and methods

**Key resources table**

| Reagent type (species) or resource | Designation | Source or reference | Identifiers | Additional information |
|---|---|---|---|---|
| Gene (*Mus musculus*) | *Vsx2* | | Ensembl: ENSMUSG00000021239 | |
| Strain, strain background (*Mus musculus.* male and female) | CD-1 | Charles River | Strain code 022 | |
| Genetic reagent (*Mus musculus*) | *Otx2*-GFP | *Fossat et al., 2007* | | |
| Genetic reagent (*Mus musculus*) | *Grm6*-YFP | *Morgan et al., 2011* | | |
| Genetic reagent (*Mus musculus*) | *Otx2*-GFP;*Grm6*-YFP | This paper | | Cross between *Otx2*-GFP and *Grm6*-YFP |
| Recombinant DNA reagent | pCAGGS | *Hsiau et al., 2007* | | Plasmid used to create *Gnb3* reporters. |
| Recombinant DNA reagent | *Gnb3*-WT-DsRed | This paper | | *Gnb3* promoter region driving DsRed |
| Recombinant DNA reagent | *Gnb3*-WT-EGFP | This paper | | *Gnb3* promoter region driving EGFP |
| Recombinant DNA reagent | *Gnb3*- K50 #1-null-DsRed- | This paper | | |
| Recombinant DNA reagent | *Gnb3*- K50 #2-null-DsRed- | This paper | | |
| Recombinant DNA reagent | *Gnb3*- K50 #3-null-DsRed- | This paper | | |
| Recombinant DNA reagent | *Gnb3*- K50 #4-null-DsRed- | This paper | | |
| Recombinant DNA reagent | *Gnb3*- K50 #5-null-DsRed- | This paper | | |

*Continued on next page*

*Continued*

| Reagent type (species) or resource | Designation | Source or reference | Identifiers | Additional information |
|---|---|---|---|---|
| Recombinant DNA reagent | *Gnb3*- K50 #3 and #5-null-DsRed- | This paper | | |
| Recombinant DNA reagent | *Gnb3*- Q50 #1-EGFP | This paper | | |
| Recombinant DNA reagent | *Gnb3*- Q50 #3-EGFP | This paper | | |
| Recombinant DNA reagent | *Gnb3*- Q50 #1 and #3-EGFP | This paper | | |
| Commercial assay or kit | Qiagen Mini-elute PCR Purification kit | Qiagen | Cat no. 28004 | |
| Commercial assay or kit | KAPA Library Quantification Kit | Roche | Cat no. 07960140001 | |
| Commercial assay or kit | SMARTer Ultra Low RNA kit for Illumina Sequencing-HV | Clonetech | Cat. Nos. 634820, 634823, 634826, 634828 and 634830) | Utilized by Washington University Genome Technology Access Core (GTAC) |
| Sequence-based reagent | Primers | IDT | | Listed in *Supplementary file 2* |
| Software, algorithm | R | http://rstudio.com | R programming language | |
| Software, algorithm | DESeq2 | https://bioconductor.org/packages/release/bioc/html/DESeq2.html | Differential gene expression analysis based on the negative binomial distribution | |
| Software, algorithm | HOMER | http://homer.ucsd.edu/homer/index.html | Hypergeometric Optimization of Motif EnRichment | |
| Software, algorithm | SAMtools | http://www.htslib.org | Samtools | |
| Software, algorithm | Bowtie2 | http://bowtie-bio.sourceforge.net/bowtie2/index.shtml | Bowtie2 | |
| Software, algorithm | Picard | https://broadinstitute.github.io/picard/ | Picard | |
| Software, algorithm | MACS2 | https://pypi.org/project/MACS2/2.1.1.20160309/ | Model-based Analysis of ChIP-Seq | |
| Software, algorithm | STAR | https://github.com/alexdobin/STAR | Spliced Transcripts Alignment to a Reference | |
| Software, algorithm | HTSeq | https://pypi.org/project/HTSeq/ | HTSeq | |

## Mouse models

All animal experiments were carried out in accordance with the regulations of the IACUC at Washington University in St. Louis. Retinal dissociation and FACS were carried out using *Otx2*-GFP or *Otx2*-GFP; *Grm6*-YFP mice. *Otx2*-GFP mice were heterozygous for a GFP cassette inserted at the C-terminus of the endogenous *Otx2* locus (*Fossat et al., 2007*). The Grm6-YFP line harbors a YFP transgene driven by the *Grm6* promoter (*Morgan et al., 2011*). All electroporation experiments were carried out in CD-1 mice.

## Retinal dissociation and FACS

Following dissection, retinas of mice aged 6–8 weeks, or 3 months (for one biological replicate of *Figure 1—figure supplement 2*) were dissociated with papain as described previously (*Trimarchi et al., 2007*). Briefly, two retinas were incubated in 400 µl of calcium/magnesium free Hanks' Balanced Salt Solution (HBSS) (Thermo Fisher) containing 0.65 mg papain (Worthington

Biochem) for 10 min at 37°C. Cells were then washed in a DMEM (Thermo Fisher) solution containing 100 units DNase1 (Roche) and incubated an additional 5 min at 37°C. Cells were then resuspended in 600 µl of sorting buffer (2.5 mM EDTA, 25 mM HEPES, 1% BSA in HBSS) and used directly for sorting. Cells were sorted on a FACS Aria-II (BD biosciences) with gates based on forward scatter, side scatter, and GFP fluorescence. OFF bipolar cell populations (*Otx2*-GFP$^+$;*Grm6*-YFP$^-$) were immediately sorted a second time to increase purity.

## Generation of reporter constructs

An 820 bp region encompassing part of the *Gnb3* 5' UTR and upstream sequence was amplified from mouse genomic DNA. Site-directed mutagenesis by overlap extension was used to modify K50 sites (*Ho et al., 1989*). The resultant PCR products were digested and ligated into GFP or DsRed reporter vectors derived from pCAGGS (*Hsiau et al., 2007*). After verification by sequencing, plasmid DNA was resuspended in PBS at a concentration of ~ 6–7 µg/µl prior to injection. All primers are listed in *Supplementary file 2*.

## In vivo and explant electroporation

In vivo subretinal injection and electroporation of newborn CD1 mice was performed as previously described (*Matsuda and Cepko, 2004*). Briefly, mice were first anesthetized on ice. A 30-gauge needle was then used to incise the eyelid and puncture the sclera, and a Hamilton syringe with a 33-gauge blunt-tipped needle was used to inject the DNA into the subretinal space. Tweezer electrodes placed across the head were then used to electroporate with five square pulses of 80 volts and 50 millisecond duration at 950 millisecond intervals. Explant electroporation was carried out as described previously (*Hsiau et al., 2007*), except that the electroporation chamber contained a solution of 0.5 µg/µl DNA in PBS.

## Retinal tissue sectioning and imaging

Eyes were removed at P21, punctured with a 26-gauge needle, and incubated in 4% paraformaldehyde for 5 min before dissection to remove the cornea. The lens was removed, and eyes were then incubated for an additional 45 min in 4% paraformaldehyde. Eye cups were next washed in PBS and incubated overnight at 4°C in 30% sucrose-PBS. The following day, eye cups were incubated in a 1:1 mixture of OCT compound (Sakura) and sucrose-PBS before being flash frozen in OCT and stored at −80°C. Retinal sections of 14 µm were cut using a cryostat (Leica CryoCut 1800), mounted on Superfrost Plus slides (Fisher), and stored at −20°C. Prior to placement of cover slips, slides were washed with PBS to remove OCT. The sections were then stained with DAPI, and coverslips were mounted using Vectashield mounting medium (Vectorlabs). Retinal sections were imaged on a Zeiss 880 laserscanning confocal microscopy in the Washington University Center for Cellular Imaging (WUCCI) Core.

## ATAC-seq

Transposition and library preparation from sorted cell populations were performed as previously described (*Buenrostro et al., 2015*). Briefly, 30,000–100,000 sorted cells were pelleted at 500 G and washed twice in ice-cold PBS before lysis. Transposition reactions were incubated at 37°C for 30 min and purified using a Qiagen MiniElute PCR Purification kit. Libraries were amplified with Phusion High-Fidelity DNA Polymerase (NEB). Cycle number was calibrated by a parallel qPCR reaction. Gel electrophoresis was used to assess library quality, and final libraries were quantified using KAPA Library Quantification Kit (KAPA Biosystems). Equimolar concentrations of each library were pooled and run on an Illumina HiSeq2500 to obtain 50 bp paired-end reads.

## ATAC-seq, DNase-seq, and RNA-seq data processing

ATAC-seq and RNA-seq reads from bipolar cell populations were processed in an identical manner to those previously obtained from rod and cone photoreceptor cells (*Hughes et al., 2017*). ATAC-seq reads were aligned to the GRCm38/mm10 mouse genome assembly using Bowtie2 (v2.3.5) with a max fragment size of 2000 (*Langmead and Salzberg, 2012*). Alignments were filtered using SAMtools (v1.9) (*Li et al., 2009*), PCR duplicates were removed using Picard (v2.19.0) (https://broadinstitute.github.io/picard/), and nucleosome-free reads were selected by removing alignments with an

insertion size greater than 150 bp. Peaks were called using MACS2 (v2.1.1) (*Zhang et al., 2008*) and annotated with HOMER (v4.8) (*Heinz et al., 2010*). DNase-seq datasets generated by ENCODE (*ENCODE Project Consortium, 2012*) were downloaded as FASTQ files from https://www.encode-project.org/ and processed in the same manner as ATAC-seq data. RNA-seq reads were aligned to the GRCm38/mm10 using STAR (v2.7.0d) (*Dobin et al., 2013*), with an index prepared for 50 base-pair reads and the RefSeq gene model, and read counts were calculated using HTSeq (v1.9) (*Anders et al., 2015*). All datasets are listed in *Supplementary file 3*.

## Processing of single-cell RNA-seq data

Single-cell data from Shekhar et al. were downloaded from the Gene Expression Omnibus (GSE81904) and processed as described by the authors to yield a digital expression matrix of normalized counts for each gene for each cell as well as cluster assignments for each cell. 'Pseudo-bulk' expression estimates were then generated by taking the weighted average of counts for each gene across cells belonging to clusters that constituted bulk populations of interest (RBC, BC5A, BC5C, BC5D, BC6, BC7, and BC8/BC9 for ON BCs; BC1A, BC1B, BC2, BC3A, BC3B, BC4 for OFF BCs; and these clusters combined for pan BCs). Finally, pseudo-bulk expression estimates were re-scaled to match bulk expression estimates.

## Transcription factor binding site motif analysis

Motif enrichment, co-enrichment, and spacing analyses for ATAC-seq and DNase-seq datasets were performed as described previously using HOMER (v4.8) (*Hughes et al., 2017*; *Heinz et al., 2010*). Differential motif enrichment was determined using a test of equal proportions (R stats v3.5.3) to compare each motif between pan BC and rods, blue cones or green cones. The top motifs across the three comparisons were manually filtered for redundancy and are shown in *Figure 4*. Motif co-occurrence analysis was performed using a list of 66 non-redundant motifs (*Hughes et al., 2017*) to which the motif for LHX3 (representing a Q50 HD motif) was manually added. For purposes of the analysis, peaks from rod, green cones and blue cones were merged to obtain a 'photoreceptor' peak list, while those from pan-, ON- and OFF-bipolar cells were merged to create a 'bipolar cell' peak list. Enrichment for co-occurrence was calculated by taking the $\log_2$(observed pairs +1/ expected pairs+1). Expected frequency of individual pairs was estimated from the counts for each motif within the pair (motif one count $\times$ motif two count $\div$ number of total peaks). Differential enrichment between tissues was calculated with a Fisher's exact test. Motif spacing was analyzed for the top enriched peaks shown in *Figure 4*. The same set of peaks for co-occurrence were centered on individual K50 or Q50 motifs, and the density of flanking secondary motifs was plotted on either strand.

## Identification of differentially accessible peaks and differentially expressed genes

DESeq2 (v1.14.1) (*Love et al., 2014*) was used to test for differential expression or differential accessibility using a $\log_2$ fold-change threshold of 1 and an FDR of 0.05. For comparison of ATAC-seq data with DNase-seq data from non-retinal tissues (*Figure 5*), photoreceptors were collapsed into a single level. Differentially expressed genes are listed in *Supplementary file 5*, and differentially accessible regions are listed in *Supplementary file 7*. For each comparison (i.e. ON versus OFF, pan BC versus rod), gene expression stemming from low-level contamination of bipolar cell populations with either rod or cone photoreceptors was filtered out. When comparing pan BC to photoreceptor populations, potential contaminating genes from the alternate photoreceptor type (i.e. rod genes identified as enriched in pan BC versus blue cone) were identified as those highly expressed (>16 fold) and specific to rod compared to blue cone, and also more highly expressed in rod compared to pan BC (at least four-fold). In comparing ON and OFF bipolar cells, genes enriched in each bipolar cell population were filtered for those which were also identified as highly specific to either photoreceptor population compared to the enriched bipolar cell type. For example, genes increased in OFF- compared to ON-bipolar cells were filtered for genes that were also highly enriched (>16 fold) in rods and blue cones compared to OFF bipolar cells. In total, 38 genes were filtered from those identified as enriched in pan BC compared to photoreceptors, and 39 genes were filtered from the ON versus OFF comparison (12 from the ON-enriched, 27 from the OFF-enriched). These genes

include those known to be expressed at very high levels in either rod or cone photoreceptors (*Corbo et al., 2007*).

## RNA isolation, qPCR and RNA-seq

Sorted bipolar cell populations were resuspended in 500 µl TRIzol reagent (Invitrogen), and RNA was isolated according to manufacturer's instructions. Prior to sequencing, RNA quality was analyzed using an Agilent Bioanalyzer. cDNA was prepared using the SMARTer Ultra Low RNA kit for Illumina Sequencing-HV (Clontech) per manufacturer's instructions. cDNA was fragmented using a Covaris E210 sonicator using duty cycle 10, intensity 5, cycles/burst 200, time 180 s. cDNA was blunt-ended, an 'A' base added to the 3′ ends, and Illumina sequencing adapters were ligated to the ends of the cDNAs. Ligated fragments were amplified for 12 cycles using primers incorporating unique index tags. Replicate libraries from each bipolar cell population were pooled in equimolar ratios and sequenced on an Illumina HiSeq 3000 (single-end 50 bp reads). For qPCR, RNA samples were treated with TURBO DNase (Invitrogen) and cDNA was synthesized with SuperScript IV (Invitrogen) and oligo(dT) primers according to manufacturer's instructions. For *Figure 1C*, expression was normalized to the average of reference genes *Gapdh*, *Sdha*, *Hprt*, and *Pgk*. For *Figure 1—figure supplement 2*, expression was normalized to *Gapdh* alone. Primers for *Grm6*, *Gnat1*, *Lhx1*, *Pax6*, *Rlbp1*, *Slc17a6*, *Vsx2*, *Grik1*, *Tacr3*, *Isl1*, and *Lrrtm1* are listed in *Supplementary file 2*.

## Acknowledgements

The authors would like to thank Leo Volkov and Yohey Ogawa for critical reading of the manuscript. The *Otx2*-GFP mouse line was a generous gift from Dr. Thomas Lamonerie (Université Côte d'Azur), and the *Grm6*-YFP line was a generous gift from Dr. Daniel Kerschensteiner (Washington University). We also credit the ENCODE consortium for the DNase-seq datasets, the Genome Technology Access Core (GTAC) in the Department of Genetics at Washington University in St. Louis for next-generation sequencing, and the Flow Cytometry Core in the Department of Pathology and Immunology at Washington University in Saint Louis for FACS services. This work was supported by the National Institute of Health (EY025196, EY026672, and EY024958 to JCC and T32EY013360 and F32EY029571 to DPM)

## Additional information

### Funding

| Funder | Grant reference number | Author |
| --- | --- | --- |
| National Eye Institute | F32EY029571 | Daniel P Murphy |
| National Eye Institute | T32EY013360 | Daniel P Murphy |
| National Eye Institute | R01EY025196 | Joseph C Corbo |
| National Eye Institute | R01EY026672 | Joseph C Corbo |
| National Eye Institute | R01EY024958 | Joseph C Corbo |

The funders had no role in study design, data collection and interpretation, or the decision to submit the work for publication.

### Author contributions

Daniel P Murphy, Conceptualization, Data curation, Formal analysis, Funding acquisition, Investigation, Methodology, Writing—original draft, Writing—review and editing; Andrew EO Hughes, Data curation, Formal analysis, Validation, Investigation, Visualization, Methodology, Writing—review and editing; Karen A Lawrence, Formal analysis, Investigation; Connie A Myers, Formal analysis, Investigation, Methodology, Writing—review and editing; Joseph C Corbo, Conceptualization, Supervision, Funding acquisition, Investigation, Writing—original draft, Project administration, Writing—review and editing

**Author ORCIDs**
Daniel P Murphy (iD) https://orcid.org/0000-0003-4846-4608
Joseph C Corbo (iD) https://orcid.org/0000-0002-9323-7140

**Ethics**
Animal experimentation: All animal experiments were carried out in accordance with the regulations of the Institutional Animal Care and Use Committee (IACUC) at Washington University in St. Louis (protocol 20170054).

**Decision letter and Author response**
Decision letter https://doi.org/10.7554/eLife.48216.043
Author response https://doi.org/10.7554/eLife.48216.044

# Additional files

### Supplementary files

• Supplementary file 1. Biological replicate and sequencing metrics for ATAC-seq and RNA-seq. 'Raw sequencing reads' are the number of paired reads for each sample. 'Processed reads' are those reads remaining after filtering out those that are improperly paired, have poor mapping quality, align to the mitochondrial genome, align to ENCODE blacklisted regions, or arise from PCR duplicates. RIN = RNA integrity number.
DOI: https://doi.org/10.7554/eLife.48216.019

• Supplementary file 2. Primers used in this work. Primers used in creation of *Gnb3* promoter constructs and in RT-qPCR experiments are listed.
DOI: https://doi.org/10.7554/eLife.48216.020

• Supplementary file 3. Datasets and accessions.
DOI: https://doi.org/10.7554/eLife.48216.021

• Supplementary file 4. Annotated ATAC-seq peaks and counts. Raw count data for all ATAC-seq peaks identified in photoreceptor and bipolar cell populations. Peaks identified in individual replicates from each cell type are shown on separate sheets.
DOI: https://doi.org/10.7554/eLife.48216.022

• Supplementary file 5. Differentially expressed genes. Genes identified as differentially expressed between aggregate bipolar cells and either rod or blue cone, and between ON and OFF bipolar cell populations are shown on separate sheets. 'Specificity' indicates which cell type expressed the gene more highly. For pan BC vs rod and cone, genes identified as putative transcription factors are identified by their TF family. Genes absent from the Drop-seq data shown in *Figure 2—source data 1* are indicated among those that are differentially expressed between ON and OFF bipolar cells.
DOI: https://doi.org/10.7554/eLife.48216.023

• Supplementary file 6. GO analysis of differentially expressed genes. Enriched GO terms for biological processes obtained from geneontology.org. Outputs for genes enriched in photoreceptor and bipolar cells are shown on separate sheets. Input gene lists were filtered based on fold-change in expression and minimum read counts to identify those most highly enriched in photoreceptor (n = 818) and bipolar cells (n = 832). A list of all genes identified by RNA-seq in either cell class was used as a reference.
DOI: https://doi.org/10.7554/eLife.48216.024

• Supplementary file 7. Differentially accessible regions. ATAC-seq peaks, normalized read counts, fold-change values, adjusted p-values and assigned genes are listed on separate sheets for each comparison. 'Specificity' indicates the cell type in which the peak is more highly accessible. 'Shared-unfiltered' peaks are those that are not differentially accessible when comparing bipolar cells versus photoreceptors (fold-change values < 2 and >-2). 'Retina' peaks are those shown in *Figure 5A*; they have been filtered to remove those accessible in B cells, brain and liver. Peaks with correlated gene expression identified in *Figure 5C* and *Figure 5—figure supplement 1D* are indicated.
DOI: https://doi.org/10.7554/eLife.48216.025

- Supplementary file 8. Known motifs enriched in enhancers of bipolar cell populations. Enrichment of all 319 motifs in the HOMER database for all, ON-, and OFF-bipolar cells, each on separate sheets. A comparison of the proportional enrichment for each motif between aggregate bipolar cells and rod, blue and green cones is included on a separate sheet. A complete list of sequence logos and position weight matrices for individual motifs is available online in the HOMER motif database: http://homer.salk.edu/homer/motif/HomerMotifDB/homerResults.html.
DOI: https://doi.org/10.7554/eLife.48216.026

- Transparent reporting form  DOI: https://doi.org/10.7554/eLife.48216.027

## Data availability

Sequencing data have been deposited in GEO under accession code GSE131625.

The following dataset was generated:

| Author(s) | Year | Dataset title | Dataset URL | Database and Identifier |
|---|---|---|---|---|
| Murphy DP, Hughes AE, Lawrence KA, Myers CA, Corbo JC | 2019 | Cis-regulatory basis of sister cell type divergence in the vertebrate retina | https://www.ncbi.nlm.nih.gov/geo/query/acc.cgi?acc=GSE131625 | NCBI Gene Expression Omnibus, GSE131625 |

The following previously published datasets were used:

| Author(s) | Year | Dataset title | Dataset URL | Database and Identifier |
|---|---|---|---|---|
| Hughes AE, Enright JM, Myers CA, Shen SQ, Corbo JC | 2017 | ATAC-seq and RNA-seq of adult mouse rods and cones | https://www.ncbi.nlm.nih.gov/geo/query/acc.cgi?acc=GSE83312 | NCBI Gene Expression Omnibus, GSE83312 |
| John Stamatoyannopoulos | 2012 | DNase on 8 week adult mouse retina | https://www.encodeproject.org/experiments/ENCSR000CNW/ | ENCODE, ENCSR000CNW |
| John Stamatoyannopoulos | 2011 | DNase-seq and DGF on 8 week mouse whole brain | https://www.encodeproject.org/experiments/ENCSR000COF/ | ENCODE, ENCSR000COF |
| John Stamatoyannopoulos | 2011 | DNase-seq on 8 week mouse liver | https://www.encodeproject.org/experiments/ENCSR000CNI/ | ENCODE, ENCSR000CNI |
| John Stamatoyannopoulos | 2011 | DNase-seq and DGF on 8 week mouse B-cell (CD43-) | https://www.encodeproject.org/experiments/ENCSR000CMN/ | ENCODE, ENCSR000CMN |
| John Stamatoyannopoulos | 2012 | DNase-seq and DGF on 8 week adult mouse heart | https://www.encodeproject.org/experiments/ENCSR000CNE/ | ENCODE, ENCSR000CNE |

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
