## [Decision Letter]

Thank you for submitting your article "Cis-regulatory basis of sister cell type divergence in the vertebrate retina" for consideration by *eLife*. Your article has been reviewed by three peer reviewers, and the evaluation has been overseen by a Reviewing Editor and Patricia Wittkopp as the Senior Editor. The reviewers have opted to remain anonymous.

The paper has been reviewed by three excellent reviewers who have entered into an intense discussion about its strength and limitations.

As you will be able to see below, the reviewers find the comparison in transcription factor binding site usage between photoreceptors and bipolar cells to be of significant interest and worth publishing. However, the reviewers were also concerned about potential problems with the profiling of ON vs. OFF bipolar cells and the impact of the paper might depend on how well you can separate the ON and OFF bipolar cells. Only the *Grm6*-GFP, but not the YFP version that you have used, labels the ON BCs. If the data indeed are based on mixed ON/OFF populations, then this needs to be clearly stated in the manuscript that currently seems to implicate that the separation is perfect.

One suggestion would be to remove the entire ON vs. OFF portion of the manuscript, but in this case, the paper would be significantly weakened unless the comparison of photoreceptor to bipolar could be enhanced. The other alternative might be to combine the Shekhar et al. single cell data into ON and OFF pools and then compare them against your RNA-seq data to see if they are well correlated. If this were to be feasible, and the data were to support the separation, this would significantly strengthen the model. As for now, the comparison in Figure 2 is neither quantitative nor visually clear.

Reviewer #1:

In order to compare the cis-regulatory features of vertebrate photoreceptors and bipolar cells, Murphy et al. analyze the transcriptomes and open chromatin landscapes of these sister cell types. The authors find that the transcriptomes are very different, while there are similarities in the usage of cis-regulatory motifs, especially K50 motifs. Interestingly, bipolar cell-specific enhancers exhibit an enrichment of E-box motifs that are bound by bHLH TFs, while photoreceptor-specific enhancers exhibit an enrichment of Q50 motifs that are bound by Q50 homeodomain TFs. Murphy et al. demonstrate that a reporter, which labels both photoreceptors and bipolar cells, can be made photoreceptor-specific by mutating its K50 to Q50 motifs. They propose that such conversion and partitioning of cis-regulatory motifs, together with the emergence of transcriptional repressors (such as VSX2 that binds to Q50 motifs), were critical steps in the evolutionary divergence of photoreceptors and bipolar cells as separate sister cell types.

This rigorous and well-written manuscript should be of interest for a broad readership, as it proposes an exciting model for how closely related cell classes are distinguished during evolution and provides a great resource for the retina community.

- Is there anything special concerning the expression and/or function subset of genes that use dimeric Q50 motifs (Figure 4)? Do they have anything else in common?

- In the first paragraph of the Discussion section: “Thus, in both retina and spinal cord, expression of VSX2 promotes interneuron fate at the expense of the alternative 'effector' neuron (photoreceptor or motor neuron) cell type.” The wording is confusing, as a photoreceptor is not an effector neuron.

Reviewer #2:

In their manuscript, Murphy et al. describe a study which combines expression (with RNA-seq) and open chromatin (with ATAC-seq) profiling of the ON-bipolar (BC) and OFF-BC cells in the mouse retina. The aim is to identify TF binding motifs that function to differentiate the bipolar cells and the photoreceptors (PRs) and then ON-BCs from OFF-BCs. I have very mixed feelings about this paper for reasons outlined below.

Authors performed RNA-seq and ATAC-seq on cells FACS-sorted based on *Otx2*-GFP and *Grm6*-YFP to obtain BCs and to subdivide them into ON and OFF subclasses (*Otx2*-GFP/*Grm6*-YFP +/+ and +/- respectively, that can be discriminated based on brightness). However, according to Schubert et al. (2008), which describes the *Grm6*-YFP mice, "YFP is expressed in a subset of on-bipolar cells", i.e. sparsely, which allows them to do electrophysiology on these cells. If I read this correctly, this means that *Otx2*-GFP^+^/*Grm6*-YFP^-^ population should contain a significant number of ON-BCs which did not express *Grm6*-YFP. Why *Grm6* RT-PCR is clean is not clear, but other markers should also be used to rule out that ON is contaminating OFF population. Next, Shekhar et al. (2016) also found that one of the ON-BC subtypes (BC5D) expresses *Grm6* at low levels if at all, probably resulting in little or no YFP expression and thus these cells would sort with the OFF population. Grm6 is expressed in the Rod BCs (RBCs) which authors here group with ON-BCs for their analysis. Yet, according to Shekhar et al. transcriptomicly ON- and OFF-BCs are more closely related to each other than ON-BCs are related to RBCs. This is pretty obvious from Figure 2, left panel. All of these issues have a strong potential to muddle the ON vs. OFF comparisons and should be addressed by the authors.

Indeed, perhaps the above issues are the reason why authors find that the relationship between chromatin accessibility and gene expression in the ON- and OFF-BCs is "complex". Yet in Figure 5—figure supplement 1C these are pretty clearly negatively correlated, a puzzling result. It would be nice to see this shown as in Figure 5C. Furthermore, this negative correlation would be interesting to explore for both, the BC vs. PRs as well as ON vs. OFF BCs. Thus, to me it seems that the strategy used in the second to last section of the Results (subsection “Photoreceptor- and bipolar-specific open chromatin regions are positively correlated with cell class-specific gene expression”) should be also applied to the ON vs. OFF, and also perhaps to be modified for the anti-correlated genes.

It is not clear to me why the authors did not use the single cell expression data from Shekhar et al. instead of their own RNA-seq data for comparison with the accessible chromatin regions. (By the way, authors' point that their pooled RNA-seq is somewhat more sensitive is a bit hollow given Shekhar et al. effort to optimize the tradeoff between the sequencing depth and the sequenced cell number.) Would it be possible to classify genes based on their expression in a particular cell type (or cell type pool) and to use the accessibility information simply to know where to look for the TF binding sites (i.e. without worrying about the enrichment of the accessibility)? If this were possible, authors could try to go deeper down the branches of the transcriptional relatedness tree from Shekhar et al., then just the first ON vs. OFF branch. A related question to this is whether (and how frequently) authors found different accessible chromatin regions for the same gene that were enriched in different cell classes?

I would remove Figure 2 (at least the right side) and Figure 2—figure supplement 2. They seem to be just reprinted Shekhar et al. data reordered according to RNA-seq "lowest adjusted p-value" of the present study. If authors want to compare the gene expression from the two studies, they need to find a way to pool Shekhar et al. data into PR, ON and OFF classes.

Regarding the electroporation experiments: While it is a nice idea to replace useless Q50 sites with K50 and show transcriptional repression, this experiment does not address the hypothesis posed in the second paragraph of the subsection “Photoreceptor and bipolar cells employ closely related yet distinct cis-regulatory grammars”, i.e. it does not show whether the presence of K50 makes chromatin inaccessible. Also, in the rest of the paper a distinction is made between enhancers and promoters, and the analysis of TF binding sites was done with enhancers, yet the electroporation experiment was done with a promoter (by their definition of <1000bp from a transcription start site) and in fact raises doubt whether the enhancer/promoter partition used by the authors in their analysis was the biologically relevant one. Finally, this experiment relies on pairing of two different TF binding sites, while sequence analysis showed that there is no enrichment of specific pairs in PR vs. BC comparison. In fact, it is not obvious to me why the authors did not find differential enrichment of the TF binding site pairs where one site is a differentially enriched TF binding site (Q50 and E-box).

In the Discussion, the evolutionary hypothesis is a nice touch, however, given that there are no experiments actually addressing it, does it deserve such a prominent role? Meanwhile, the many interesting observations (e.g. why there is no enrichment of TF binding pairs) remain unaddressed.

Finally, a major concern is that the paper is difficult to follow and was not written with a broad readership in mind. The authors reasoning is often not clear (e.g. subsection “Bipolar cells have a more accessible chromatin landscape than either rods or cones”, first paragraph), the definitions of the terms used are hard to find (promoter vs. enhancer is defined in the legend of a figure supplement, the fact that they use "TSS-distal" and "enhancer" as synonyms is also hidden in figure legends), figure legends are not clear (e.g. Figure 4—figure supplement 1) and sometimes it is just not clear what authors mean (what are "replicates" in the first sentence of Figure 3A legend). Finally, the entire paper needs some re-organization (e.g. the way PR vs. BC and ON vs. OFF comparisons are written up should parallel each other).

To sum up, while I am not against the publication of at least part of this data in *eLife*, the paper would need a major overhaul. There are seemingly large issues to address and the analysis feels superficial.

Reviewer #3:

In this manuscript, Murphy and colleagues examine the regulatory logic governing photoreceptor- and bipolar cell-specific gene expression. They generate a considerable amount of new data to describe the transcriptome and the open chromatin regions within bipolar cells. Murphy and colleagues then compare their results with prior data from photoreceptors. Interestingly, this revealed that photoreceptor and bipolar cell genes were enriched for different types of transcription factor binding sites. In one example, they show that subtly changing the type of binding site will restrict a broader expression pattern to be only made by photoreceptors. From their data, they then extrapolate a model to explain how bipolar cells evolved from a photoreceptor ancestral state. The experiments and data analysis in this manuscript are rigorous and compelling, but would be improved by some minor modifications. While the discussion about the evolution of photoreceptors is of high interest, the manuscript would be strengthened by additional discussion of the cis-regulatory findings. Specific recommendations are listed below:

1) A couple modifications to the data analysis and figures would improve the manuscript.

a) The authors mention that zinc finger transcription factor sites are more common in photoreceptors. However, this binding site preference data is not shown in Figure 4 (except for CTCF). Showing a zinc finger example in Figure 4 would improve the clarity of the manuscript.

b) In Figure 5C (and Figure 5—figure supplement 1D), it would help the reader to label some of the dots in these plots. For example, showing a few examples of known photoreceptor and bipolar genes. Highlighting a few discordant (black dots, e.g., *Grik1*) genes would help the reader as well.

c) In Figure 6D, the categorical format provides a nice summary of the data. However, a plot or table showing quantitative data and statistics would make a much stronger case that the type of binding site affects gene expression.

2) While the discussion about the evolution of photoreceptors and bipolar cells and Figure 7 are done well, it may be hard for readers to link the cis-regulatory data to the evolutionary model. In addition, there are some regulatory findings that are not discussed. The manuscript would benefit from a brief expansion of the Discussion section to address the cis-regulatory grammar findings in more detail. Addressing the following items would strengthen the manuscript:

a) Why is overall chromatin accessibility different between photoreceptors and bipolars? Does it impact the evolutionary model?

b) If bHLH factors are important for photoreceptor formation or maintenance, why are these cites depleted from mature photoreceptors? Are there differences between developmental and homeostatic/mature cis-regulatory networks? Are the cis-regulatory changes expected to be the same in a developmental context?

c) Why are bZIP sites (e.g., *Nirl*) strong in green cones and modest in rods (Figure 4)?

d) Why do some genes (e.g., *Grik1*) behave in a discordant fashion?

---

## [Author Response]

As you will be able to see below, the reviewers find the comparison in transcription factor binding site usage between photoreceptors and bipolar cells to be of significant interest and worth publishing. However, the reviewers were also concerned about potential problems with the profiling of ON vs. OFF bipolar cells and the impact of the paper might depend on how well you can separate the ON and OFF bipolar cells. Only the Grm6-GFP, but not the YFP version that you have used, labels the ON BCs. If the data indeed are based on mixed ON/OFF populations, then this needs to be clearly stated in the manuscript that currently seems to implicate that the separation is perfect.

See below, reviewer #2, first comment.

One suggestion would be to remove the entire ON vs. OFF portion of the manuscript, but in this case, the paper would be significantly weakened unless the comparison of photoreceptor to bipolar could be enhanced. The other alternative might be to combine the Shekhar et al. single cell data into ON and OFF pools and then compare them against your RNA-seq data to see if they are well correlated. If this were to be feasible, and the data were to support the separation, this would significantly strengthen the model. As for now, the comparison in Figure 2 is neither quantitative nor visually clear.

See below reviewer #2, tenth comment.

Reviewer #1:[…]- Is there anything special concerning the expression and/or function subset of genes that use dimeric Q50 motifs (Figure 4)? Do they have anything else in common?

We assume that reviewer #1 is inquiring about differences between genes which are in proximity to dimeric Q50 motifs, as opposed to those associated with monomeric Q50 motifs. To address this question, we compared lists of genes associated with peaks identified in Figure 5A (open in photoreceptor or bipolar cells, but not other tissues) which harbored either monomeric or dimeric Q50 sites, using the Gene Ontology (GO) tool PANTHER. The list of genes associated with monomeric Q50 motifs (n = 1,370) was enriched for GO biological processes relating to neuronal development, including axon guidance and dopaminergic neuronal differentiation. The list of genes associated with dimeric motifs (n = 423) was similar, but associated with fewer and more generic GO terms, including amino acid import, cell-cell adhesion, and generation of neurons. This limited analysis does not suggest that dimeric and monomeric motifs distinguish functionally distinct gene expression programs. Whether or not dimeric and monomeric Q50 sites function differently as cis-regulatory elements would require a systematic comparison of their activity, which is beyond the scope of the present study. Previous research in our lab indicates that dimeric K50 sites act as stronger activators than monomeric sites in photoreceptors (Hughes et al., 2018). Thus, it could be possible that, in bipolar cells, dimeric and monomeric Q50 sites have similar function, with dimeric motifs representing stronger repressive elements.

- In the first paragraph of the Discussion section: “Thus, in both retina and spinal cord, expression of VSX2 promotes interneuron fate at the expense of the alternative 'effector' neuron (photoreceptor or motor neuron) cell type.” The wording is confusing, as a photoreceptor is not an effector neuron.

We have removed the expression “effector neuron”.

Reviewer #2:In their manuscript, Murphy et al. describe a study which combines expression (with RNA-seq) and open chromatin (with ATAC-seq) profiling of the ON-bipolar (BC) and OFF-BC cells in the mouse retina. The aim is to identify TF binding motifs that function to differentiate the bipolar cells and the photoreceptors (PRs) and then ON-BCs from OFF-BCs. I have very mixed feelings about this paper for reasons outlined below.Authors performed RNA-seq and ATAC-seq on cells FACS-sorted based on Otx2-GFP and Grm6-YFP to obtain BCs and to subdivide them into ON and OFF subclasses (Otx2-GFP/Grm6-YFP +/+ and +/- respectively, that can be discriminated based on brightness). However, according to Schubert et al. (2008), which describes the Grm6-YFP mice, "YFP is expressed in a subset of on-bipolar cells", i.e. sparsely, which allows them to do electrophysiology on these cells. If I read this correctly, this means that Otx2-GFP^+^/Grm6-YFP^-^ population should contain a significant number of ON-BCs which did not express Grm6-YFP. Why Grm6 RT-PCR is clean is not clear, but other markers should also be used to rule out that ON is contaminating OFF population.

The reviewer’s reasoning is sound but is based on a citation error on our part. The correct citation for the source of the *Grm6*-YFP mouse line used in our study is Morgan et al. (2011). The line was referred to as ‘Grm6-YFP’ in that study and was also used in a subsequent study (Johnson et al., 2017, Nat. Commun, 8: 1220) where it was referred to as ‘Grm6_L_-YFP-DTA^con^’. This mouse line expresses YFP in the vast majority of ON bipolar cells and not in OFF bipolar cells. We have corrected the citation in the text and address the purity of our ON and OFF bipolar cell populations in the following points:

1) Co-labeling of *Grm6*-YFP positive cells with a marker for photoreceptor and bipolar cell axon terminals (VGLUT1) shows that *Grm6*-YFP positive cells terminate exclusively in the inner half of the inner plexiform layer (S3-S5), as expected of ON bipolar cells (Figure 3A, Johnson et al. 2017 Nat. Commun, 8: 1220). *Grm6*-YFP positive cells in the innermost sublamina of the IPL are also labelled with PKCα, a marker for rod bipolar cells.

2) RNA-seq read counts for pan-ON BC markers (e.g., *Grm6* and *Isl1*) are very high in the ON bipolar population and very low in the OFF bipolar population.

3) Similarly, qPCR for *Grm6* (Figure 1C) and *Isl1* (new RT-qPCR data, presented in Figure 1—figure supplement 2) show high expression in the ON bipolar population and little or no expression in the OFF bipolar population.

4) In a new analysis (presented in Figure 2—figure supplement 1) we combined single-cell RNA-seq data from Shekhar et al. to create ‘pseudo-bulk’ ON and OFF bipolar profiles for comparison to our data. We found that our RNA-seq data for ON and OFF populations correlates very well with pooled pseudo-bulk data for ON and OFF bipolar cells from Shekhar et al. In addition, we found a strong correlation between estimates of fold-change in pseudo-bulk single-cell RNA-seq data and fold-changes identified in our ON vs. OFF comparison (See response to the tenth comment below).

5) Additional analysis (see next point) indicates that a rare ON BC subtype, BC5D, is absent from both our ON and OFF populations.

Next, Shekhar et al. (2016) also found that one of the ON-BC subtypes (BC5D) expresses Grm6 at low levels if at all, probably resulting in little or no YFP expression and thus these cells would sort with the OFF population.

We thank reviewer #2 for highlighting the problem of low *Grm6* expression in BC5D cells. To determine whether BC5D cells were sorted into ON or OFF populations, we first examined the expression of the BC5D-specific marker, *Lrrtm1* (identified in Shekhar et al. 2016) in our RNA-seq data. We found that there were very few *Lrrtm1* sequence reads (<100) in either ON or OFF bipolar cells, suggesting that BC5D cells were absent from both populations. Given the low-level expression of *Grm6* in BC5D cells, we reasoned that these cells may have fallen between the fluorescence gates used in our original FACS analysis. To evaluate this possibility, we repeated our FACS analysis, but this time we also collected cells with intermediate levels of green fluorescence which fell between the previously used ON and OFF population gates (this intermediate population is termed ‘YFP-low’). We found that the YFP-low population expressed both the ON BC marker *Isl1* and the BC5D marker *Lrrtm1*, while the newly sorted ON and OFF populations again failed to express *Lrrtm1*. These results (included in Figure 1—figure supplement 2) indicate that BC5D cells were likely excluded from both ON and OFF populations in our original analysis.

We believe that the absence of BC5D cells does not in any way alter the conclusions of our study. Published estimates of the prevalence of BC5D cells suggest that this cell type comprises only a small percentage of bipolar cells in the mouse retina. Shekhar et al. found that BC5D made up only 2.3% of bipolar cells analyzed by Drop-seq. Furthermore, the authors note that BC5D cells resemble the bipolar cell type identified by electron microscopy as “XBC” by Helmstaedter et al. (2013, Nature, 500:168–174). In that publication, XBC cells comprised only ~1.5% (7 out of 459 cells) of bipolar cells analyzed. We attempted to estimate the proportion of BC5D among ON bipolar cells by quantifying the number of cells in Shekhar et al. Supplementary Figure 7F, which shows a retinal whole-mount preparation labelled with *Lrrtm1* and *Grm6* by double FISH. Using ImageJ to identify cells by particle analysis, we counted a total of 258 *Grm6* positive cells in Figure 7F, indicating that the seven *Lrrtm1* positive cells constitute about 2.7% of ON bipolar cells. While quantifying a single field of view is by no means definitive, this number correlates well with the overall estimates of BC5D representing 1.5-2.3% of all bipolar cells. We have added a discussion of these issues to the main text (subsection “Photoreceptors and bipolar cells exhibit divergent transcriptional profiles”, sixth paragraph).

Grm6 is expressed in the Rod BCs (RBCs) which authors here group with ON-BCs for their analysis. Yet, according to Shekhar et al. transcriptomicly ON- and OFF-BCs are more closely related to each other than ON-BCs are related to RBCs. This is pretty obvious from Figure 2, left panel. All of these issues have a strong potential to muddle the ON vs. OFF comparisons and should be addressed by the authors.

The grouping of RBCs and cone ON bipolar cells is a drawback of our sorting approach. Our analysis nonetheless suggests that there are very few differences in chromatin accessibility between individual BC types. We estimate that RBCs constitute nearly 60% of the cells in our ON population (RBCs compose 56% of ON bipolar cells identified in Shekhar et al. and ~57% of those identified as ON BCs by Wässle et al., 2009). Thus, we predict that RNA-seq and ATAC-seq data from isolated rod bipolar cells will likely resemble our ON population data quite closely. We have addressed these issues in a new paragraph in the Discussion.

Indeed, perhaps the above issues are the reason why authors find that the relationship between chromatin accessibility and gene expression in the ON- and OFF-BCs is "complex". Yet in the Figure 5—figure supplement 1C these are pretty clearly negatively correlated, a puzzling result. It would be nice to see this shown as in Figure 5C. Furthermore, this negative correlation would be interesting to explore for both, the BC vs. PRs as well as ON vs. OFF BCs.

We thank the reviewer for pointing this out. We inadvertently transposed the X-axis labels of Figure 5—figure supplement 1 in the original submission. We have corrected this error. In addition, we now include a comparison between ON and OFF bipolar cells as suggested by the reviewer (see Figure 5—figure supplement 1).

Thus, to me it seems that the strategy used in the second to last section of the Results (subsection “Photoreceptor- and bipolar-specific open chromatin regions are positively correlated with cell class-specific gene expression”) should be also applied to the ON vs. OFF, and also perhaps to be modified for the anti-correlated genes.

This is a great suggestion. We attempted to remove peaks open in other tissues from the ON vs. OFF comparison, as was done for the photoreceptor/bipolar cell comparison. However, this yielded a very small number of remaining peaks which are more accessible in OFF bipolar cells than in ON (n = 25). We think this is largely a function of the already small number of regions more accessible in OFF bipolar cells and highlights the high degree of similarity between the two datasets.

It is not clear to me why the authors did not use the single cell expression data from Shekhar et al. instead of their own RNA-seq data for comparison with the accessible chromatin regions.

Our goal was to compare chromatin accessibility and transcriptomic data from identical bipolar cell populations. We therefore performed both analyses on the same populations isolated in an identical manner.

(By the way, authors' point that their pooled RNA-seq is somewhat more sensitive is a bit hollow given Shekhar et al. effort to optimize the tradeoff between the sequencing depth and the sequenced cell number.)

As described by Shekhar et al., the authors optimized their experimental approach for cost-effective discovery and classification of BC subtypes. Single-cell RNA-seq proved to be a powerful approach for unsupervised discovery of BC subtypes, characterization of BC subtype transcriptomes, and identification of BC subtype marker genes. Consistent with our results, previous comparisons of single-cell and bulk RNA-sequencing have suggested that bulk sequencing has higher sensitivity to detect (and precision to quantify) the transcription of genes with low expression. Therefore, we believe bulk RNA-seq from purified populations of BCs provides a complementary resource for understanding BC gene expression over a broad dynamic range. In response to this comment, we have modified the relevant section of the text (subsection “Photoreceptors and bipolar cells exhibit divergent transcriptional profiles”, seventh paragraph).

Would it be possible to classify genes based on their expression in a particular cell type (or cell type pool) and to use the accessibility information simply to know where to look for the TF binding sites (i.e. without worrying about the enrichment of the accessibility)? If this were possible, authors could try to go deeper down the branches of the transcriptional relatedness tree from Shekhar et al., then just the first ON vs. OFF branch.

This is an excellent suggestion, and we have performed the analysis. Unfortunately, we did not identify convincing differences in motif content between peak sets associated with different subtypes of bipolar cells. We evaluated several different strategies for associating genes identified as cluster-enriched by Shekharet al. with ATAC-seq peaks that we identified by profiling bulk populations of bipolar cells, including using peaks within 5, 10, 25, or 50 kb of the transcription start sites corresponding to cluster-enriched genes, and considering promoter and enhancer peaks separately as well as jointly. We then performed motif enrichment analysis on the resulting peak sets. We found that none of these approaches revealed strong differences in the motifs enriched among ATAC-seq peaks associated with genes from different clusters. There are several possible explanations for this result. First, the approach may not adequately enrich for cluster-specific ATAC-seq peaks (high background of shared peaks). Second, ATAC-seq from larger populations may fail to identify cluster-specific peaks in less common bipolar cell subtypes (low signal). And third, the cis-regulatory grammars of bipolar cell subtypes may be too similar to identify differences with this approach. Ultimately, ATAC-seq from purified bipolar cell subtypes and/or single-cell ATAC-seq will be required to differentiate among these possibilities.

A related question to this is whether (and how frequently) authors found different accessible chromatin regions for the same gene that were enriched in different cell classes?

A very interesting question. We certainly do find peaks near the same gene which exhibit discordant accessibility patterns. For example, using the list of 7,144 peaks identified in Figure 5A, which are either specific to photoreceptors or to bipolar cells, there are a total of 6 peaks which were associated with *Prkca*. Despite bipolar-specific expression, four of these peaks (one of which is highlighted in Figure 5C) are more highly accessible in photoreceptor cells, while two are more highly accessible in bipolar cells. Overall, about 15% (496 bipolar cell-enriched, and 574 photoreceptor enriched) of the 7,144 peaks are associated with genes that were also assigned to peaks enriched in the alternate cell type. Perhaps this percentage reflects the high relatedness of photoreceptor and bipolar cells. We hypothesize that this percentage would be somewhat lower when comparing elements that are specific to more distantly related cell types. Of course, these analyses depend on our ability to accurately match ATAC-seq peaks with genes, which is imperfect.

I would remove Figure 2 (at least the right side) and Figure 2—figure supplement 2 They seem to be just reprinted Shekhar et al. data reordered according to RNA-seq "lowest adjusted p-value" of the present study. If authors want to compare the gene expression from the two studies, they need to find a way to pool Shekhar et al. data into PR, ON and OFF classes.

We believe that it is important to highlight the correlation between our data and those of Shekhar et al. For this reason, we have chosen not to remove Figure 2 or Figure 2—figure supplement 2.

We appreciate the suggestion to pool data from Shekhar et al. for comparison with our data. Accordingly, we reprocessed the single-cell data from Shekharet al., aggregating gene-level expression estimates for pan, ON, and OFF bipolar cells by taking the weighted average of normalized counts over all cells belonging to each class (determined by their assigned cluster). We then rescaled these values to directly compare expression estimates from bulk and ‘pseudo-bulk’ RNA-seq. This analysis showed that expression estimates from bulk and single-cell RNA-seq are largely consistent (Pearson correlation coefficients ranged from 0.82 to 0.85). As expected, fewer genes were detected by single-cell RNA-seq, and these tended to have relatively low expression as estimated by bulk RNA-seq. Furthermore, this analysis suggested that our sorting strategy enriched for the desired bipolar cell populations. That is, bulk pan bipolar cell data was more strongly correlated with single-cell RNA-seq from pan bipolar cells compared to rods or cones, and bulk ON bipolar cell data was more strongly correlated with single-cell RNA-seq from ON bipolar cells compared to OFF bipolar cells (and vice versa for OFF vs. ON bipolar cells). Finally, this analysis allowed us to compare the ON vs. OFF bipolar cell differential expression analysis between bulk and single-cell RNA-seq. We found that the resulting fold-change estimates were strongly correlated (R^2^ = 0.78), again supporting the interpretation that our sorting strategy was effective in isolating enriched populations of ON and OFF bipolar cells. We now present this analysis in detail in the Results (subsection “Photoreceptors and bipolar cells exhibit divergent transcriptional profiles”, fifth paragraph), Materials and methods section, and in Figure 2—figure supplement 1.

Regarding the electroporation experiments: While it is a nice idea to replace useless Q50 sites with K50 and show transcriptional repression, this experiment does not address the hypothesis posed in the second paragraph of the subsection “Photoreceptor and bipolar cells employ closely related yet distinct cis-regulatory grammars”, i.e. it does not show whether the presence of K50 makes chromatin inaccessible.

This is a valid point. The text (subsection “Photoreceptor and bipolar cells employ closely related yet distinct cis-regulatory grammars”, second paragraph) has been adjusted accordingly.

Also, in the rest of the paper a distinction is made between enhancers and promoters, and the analysis of TF binding sites was done with enhancers, yet the electroporation experiment was done with a promoter (by their definition of <1000bp from a transcription start site) and in fact raises doubt whether the enhancer/promoter partition used by the authors in their analysis was the biologically relevant one.

This comment identifies an important contrast between our analysis of candidate regulatory regions identified by ATAC-seq and our functional studies characterizing bipolar cell regulatory activity by electroporation. Regarding the biological relevance of promoter and enhancer designations, there are a variety of well-documented differences between enhancers and promoters, including GC content (Natarajan et al., 2012,Genome Res.,9:1711-22 and Wang et al., 2012, Genome Res., 9:1798-812) patterns of transcription factor binding site enrichment (Zheng et al., 2012, Nucleic AcidsRes., 43:74-83), and cell type-specificity (Corces et al., 2016, Nat. Genet., 48:1193–203). While the -1000 bp to +100 bp definition of ‘promoter’ used for data analysis is largely consistent with the size and location of depleted nucleosomes overlapping gene promoters (see Supplementary Figure 6A in Hughes et al., 2017), its main purpose is to exclude promoter sequences from the ‘enhancer’ set. This distinction is not meant to imply that enhancers cannot occur within 1 kb of the transcription start site. We believe the mouse *Gnb3* ‘promoter region’ contains bona fide enhancer sequences.

Finally, this experiment relies on pairing of two different TF binding sites, while sequence analysis showed that there is no enrichment of specific pairs in PR vs. BC comparison. In fact, it is not obvious to me why the authors did not find differential enrichment of the TF binding site pairs where one site is a differentially enriched TF binding site (Q50 and E-box).

The reviewer is correct in expecting that individually enriched TF sites would lead to pairs which are differentially enriched between the two cell populations. This is in fact the case. Our original analysis was structured to identify combinations of TF binding sites that were enriched given the baseline frequencies of each motif in each peak set. We took this approach reasoning that it was more likely to identify pairs of motifs that function cooperatively (and that the enrichment of pairs of motifs among one peak set vs. another would be trivial if one or both had already been shown to be enriched). We have now reanalyzed motif co-occurrence without normalizing for baseline enrichment. As expected, we find that motifs which are highly enriched in each tissue (e.g. E-box in bipolar cells and homeodomain in photoreceptors) appear in motif pairs which are also enriched in the respective cell population. We have updated our supplementary figure accordingly (see Figure 4—figure supplement 1). In addition, we state the result of the original analysis: although specific pairs are enriched in one peak set vs. another, the frequencies of peaks with one or both motifs are consistent with an independent model. Please see the last paragraph of the subsection “Photoreceptor and bipolar cells employ closely related yet distinct cis-regulatory grammars”.

In the Discussion, the evolutionary hypothesis is a nice touch, however, given that there are no experiments actually addressing it, does it deserve such a prominent role? Meanwhile, the many interesting observations (e.g. why there is no enrichment of TF binding pairs) remain unaddressed.

We now address multiple additional points in the Discussion. Please see the above comment in regard to motif co-enrichment.

Finally, a major concern is that the paper is difficult to follow and was not written with a broad readership in mind. The authors reasoning is often not clear (e.g. subsection “Bipolar cells have a more accessible chromatin landscape than either rods or cones”, first paragraph), the definitions of the terms used are hard to find (promoter vs. enhancer is defined in the legend of a figure supplement, the fact that they use "TSS-distal" and "enhancer" as synonyms is also hidden in figure legends), figure legends are not clear (e.g. Figure 4—figure supplement 1) and sometimes it is just not clear what authors mean (what are "replicates" in the first sentence of Figure 3A legend). Finally, the entire paper needs some re-organization (e.g. the way PR vs. BC and ON vs. OFF comparisons are written up should parallel each other).

Thank you for bringing up these issues. We have made the following efforts to clarify the text:

- The second paragraph of the subsection “Bipolar cells have a more accessible chromatin landscape than either rods or cones” has been re-written;

- We have added a definition of ‘promoter’ and ‘enhancer’ to the main text (subsection “Bipolar cells have a more accessible chromatin landscape than either rods or cones”, first paragraph);

- We have updated the legend for Figure 4—figure supplement 1;- The wording in the legend of Figure 3 has been modified to improve clarity.Reviewer #3:[…]1) A couple modifications to the data analysis and figures would improve the manuscript.a) The authors mention that zinc finger transcription factor sites are more common in photoreceptors. However, this binding site preference data is not shown in Figure 4 (except for CTCF). Showing a zinc finger example in Figure 4 would improve the clarity of the manuscript.

We believe the reviewer is referring to the enrichment of zincfinger TFs themselves, especially in rods compared to bipolar cells. To clarify this point, we have changed ‘enrichment’ to ‘expression’. Otherwise, zincfinger motifs are in fact highly enriched in both tissues, specifically those corresponding to CTCF and BORIS (a homolog of CTCF with a similar binding site preference). The CTCF motif is actually more highly enriched in bipolar cells than in either rods or cones. Other zincfinger motifs are also differentially enriched, such as PRDM1 and PRDM9 (higher in bipolar cells) and ZNF711, ZNF467 (higher in photoreceptors). However, these motifs are not highly prevalent in the enhancers of either cell type, making density plots as in Figure 4 relatively uninformative.

b) In Figure 5C (and Figure 5—figure supplement 1D), it would help the reader to label some of the dots in these plots. For example, showing a few examples of known photoreceptor and bipolar genes. Highlighting a few discordant (black dots, e.g., Grik1) genes would help the reader as well.

Done.

c) In Figure 6D, the categorical format provides a nice summary of the data. However, a plot or table showing quantitative data and statistics would make a much stronger case that the type of binding site affects gene expression.

Unfortunately, we do not have quantitative data for this section. To demonstrate the reproducibility of our findings, we have compiled a new supplementary figure (Figure 6—figure supplement 1) with images of additional retinas electroporated with the various *Gnb3* reporter constructs.

2) While the discussion about the evolution of photoreceptors and bipolar cells and Figure 7 are done well, it may be hard for readers to link the cis-regulatory data to the evolutionary model. In addition, there are some regulatory findings that are not discussed. The manuscript would benefit from a brief expansion of the Discussion section to address the cis-regulatory grammar findings in more detail. Addressing the following items would strengthen the manuscript:a) Why is overall chromatin accessibility different between photoreceptors and bipolars? Does it impact the evolutionary model?

This finding may partly be attributable to the uniquely closed architecture of rod chromatin documented in a prior study (Hughes et al., 2017). This topic is addressed throughout the Discussion.

b) If bHLH factors are important for photoreceptor formation or maintenance, why are these cites depleted from mature photoreceptors? Are there differences between developmental and homeostatic/mature cis-regulatory networks? Are the cis-regulatory changes expected to be the same in a developmental context?

This question highlights an important distinction between the motif density data presented in Figure 4 and that presented in Figure 5. Specifically, the data from Figure 5 is derived from peak sets that are specific to either photoreceptor or bipolar cells. When all regions of accessibility are considered (as in Figure 4), both photoreceptor and bipolar cells show an overall enrichment of E-box motifs, albeit to a different degree. As such, the lack of E-box motif enrichment in the peak set specific to photoreceptors does not necessarily mean that E-box motifs are depleted from the cis-regulatory regions of mature photoreceptors. Rather, this shows that within the specific subset of peaks which are accessible in photoreceptors but not in bipolar cells or other cell types, E-box motifs are not observed in high density. The importance of E-box motifs in photoreceptor cis-regulation is indicated by the high density of E-box motifs within the set of peaks that are shared by photoreceptors and bipolar cells, but not other cell types (Figure 5C, second panel from the right). We expect that there is variation in cis-regulatory networks throughout development, and that these changes are likely to vary from cell type to cell type. It would be interesting to profile and compare the cis-regulatory landscape of developing photoreceptor and bipolar cells in future studies. We now address these points in the Results and Discussion.

c) Why are bZIP sites (e.g., Nrl) strong in green cones and modest in rods (Figure 4)?

The significance of the difference in bZIP motif enrichment in rod and cone open chromatin is an interesting and unanswered question. One possibility is that these motifs are bound by NRL (a MAF-type bZIP TF) in rods and by another currently unknown bZIP TF in cones. For example, we previously found that MAF (a MAF-type bZIP TF) is expressed in adult mouse cones (at low levels) but not in rods (see Supplementary Table 6 in Hughes et al., 2017). Additional work is needed to evaluate this idea.

d) Why do some genes (e.g., Grik1) behave in a discordant fashion?

This is an important question. Our observation that there is a correlation between accessibility and gene expression is consistent with previous reports in photoreceptors (Hughes et al., 2017) and other cell types (de la Torre-Ubieta et al., 2018 and Starks et al., 2019). However, the correlation is not perfect (i.e., accessibility is not sufficient for expression or enhancer activity). For example, the mouse LCR, an enhancer which drives expression of red/green cone opsin (*Opn1mw*), is open in both rods and cones, despite its cone-specific expression (Hughes et al., 2017). Similarly, in developmental contexts, regions that are ‘poised’ exhibit accessibility yet remain inactive until later developmental timepoints (Lara-Astiaso et al., 2014 and Schulz et al., 2015, Genome Res., 25:1715-26). Presumably, transcriptional activity in these instances requires expression of additional cell type-specific factors. We now address this in the Discussion.